# Stealing That Free Lunch:
# Exposing the Limits of Dyna-Style Reinforcement Learning

**Brett Barkley** [1]  **David Fridovich-Keil** [2]

## Abstract

Dyna-style off-policy model-based reinforcement learning (DMBRL) algorithms are a family of techniques for generating synthetic state transition data and thereby enhancing the sample efficiency of off-policy RL algorithms. This paper identifies and investigates a surprising performance gap observed when applying DMBRL algorithms across different benchmark environments with proprioceptive observations. We show that, while DMBRL algorithms perform well in control tasks in OpenAI Gym, their performance can drop significantly in DeepMind Control Suite (DMC), even though these settings offer similar tasks and identical physics backends. Modern techniques designed to address several key issues that arise in these settings do not provide a consistent improvement across all environments, and overall our results show that adding synthetic rollouts to the training process — the backbone of Dyna-style algorithms — significantly degrades performance across most DMC environments. Our findings contribute to a deeper understanding of several fundamental challenges in model-based RL and show that, like many optimization fields, there is no free lunch when evaluating performance across diverse benchmarks in RL.

## 1. Introduction

Colloquially, the "no free lunch theorem" states that no optimization algorithm can be universally optimal across all problem instances. In reinforcement learning (RL), this implies that performance will vary depending on the environment and problem characteristics. In that light, this paper begins with a simple, yet novel and unexpected observation: Model-Based Policy Optimization (MBPO) (Janner et al., 2019), a popular Dyna-style (Sutton, 1991) model-based reinforcement learning (DMBRL) algorithm, demonstrates strong performance across tasks in OpenAI Gym (Brockman et al., 2016), but performs significantly worse than its base off-policy algorithm, Soft Actor Critic (SAC) (Haarnoja et al., 2019a), when trained in DeepMind Control Suite (DMC) (Tassa et al., 2020) — cf. Figure 1.

Interestingly, the Gym and DMC benchmarks feature similar tasks and identical physics backends (Todorov et al., 2012). Yet, despite the popularity of MBPO, with around 1k citations (as of late 2024), only two studies report evaluating MBPO in DMC (Wang et al., 2024; Voelcker et al., 2024), and a performance gap has only been noted in the `hopper-hop` task (Voelcker et al., 2024). Our results generalize these findings, and call into question the robustness of MBPO, and of Dyna-style algorithms more broadly.

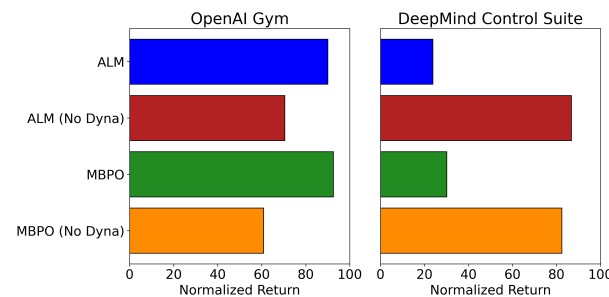

*Figure 1.* Comparison of normalized final return for two different DMBRL algorithms with and without Dyna-style enhancements. Results are averaged across 6 random seeds per task, with 6 tasks from OpenAI Gym (Brockman et al., 2016) and 15 from DMC (Tassa et al., 2020). For training curves, cf. Figures 3 and 17.

Beyond robustness, these discrepancies raise critical questions for us in the RL community. First, is the performance gap fundamental, or merely an artifact we can "fix" with modern techniques applied to predictive models, other model-based RL components, and/or the base off-policy

---

[1]Department of Computer Science, University of Texas at Austin, Austin, TX, USA [2]Department of Aerospace Engineering and Engineering Mechanics, University of Texas at Austin, Austin, TX, USA. Correspondence to: Brett Barkley <bbarkley@utexas.edu>.

*Proceedings of the $42^{nd}$ International Conference on Machine Learning*, Vancouver, Canada. PMLR 267, 2025. Copyright 2025 by the author(s).

algorithms?

Our extensive results indicate: "yes they are fundamental, and no we cannot." Figure 1 shows that this gap persists more broadly than in just MBPO, but also in another recent DMBRL algorithm, Aligned Latent Models (ALM) (Ghugare et al., 2022). These algorithms have contrasting design philosophies: MBPO is built on SAC (Haarnoja et al., 2019a), a stochastic policy algorithm leveraging entropy-based exploration, while ALM is built on DDPG (Lillicrap et al., 2019), a deterministic policy algorithm that relies on additive Gaussian noise for exploration. These differences in exploration strategies, combined with their distinct architectures and objectives, make MBPO and ALM complementary test cases for assessing whether the observed gap is inherent to Dyna-based methods rather than specific to a particular implementation. While our findings raise serious concerns about these instantiations, our intent is not to challenge the theoretical soundness of the Dyna framework itself, but rather to evaluate how well its practical realizations hold up under realistic, high-fidelity conditions.

Using MBPO as a surrogate for a subclass of DMBRL algorithms that underperform in DMC, we analyze the Gym/DMC performance gap and try to close it by addressing some possible sources of this discrepancy including overestimation bias, neural network plasticity, and environment modeling fidelity. Despite these efforts, there remains a substantial gap in performance on these two benchmarks, which suggests that the generalization challenges of MBPO — and, by extension, a subclass of DMBRL algorithms — are more deeply rooted.

Second, this oversight remaining unnoticed for several years highlights a significant, pervasive issue which has also noted in several other AI subfields (Haibe-Kains et al., 2020; Bender et al., 2021; Agarwal et al., 2021; Ahmed et al., 2023; Jordan et al., 2024): a lack of reasonable access which prevents scientists from reproducing and verifying key results. For instance, per the calculations in Appendix B, generating just Figure 4 of this manuscript using the original MBPO implementation (Janner, 2019) would require over 100 GPU-days, assuming optimistic runtime estimates from prior work (Ghugare et al., 2022; Xu et al., 2022).

Such computational barriers limit experimentation and cross-comparison, leaving critical issues, like those we highlight, unexamined. To address this, we developed a MBPO implementation built on a high-performance SAC implementation (D'Oro et al., 2023), leveraging JAX (Bradbury et al., 2018) for efficient parallelization. This new implementation reduces training time for the experiments in Figure 4 to approximately 4 GPU-days, a $26\times$ speedup compared to the original Pytorch implementation (Janner et al., 2019).

Our implementation also compares favorably to Aligned

Latent Models (ALM), a model-based algorithm known for its high wall-clock speed, as shown in Figure 2. These improvements over previous model-based RL implementations empower researchers with limited resources to conduct extensive DMBRL studies using just a single GPU.

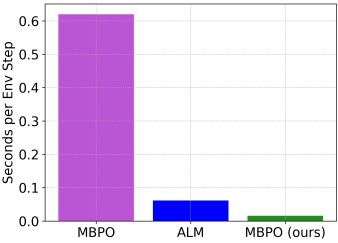

*Figure 2.* Comparison of seconds per environment step across multiple DMBRL implementations when deployed on 6 OpenAI Gym environments. Compared to Pytorch MBPO and ALM, our implementation takes $\sim 40\times$ and $\sim 4\times$ less time, respectively.

In summary, our main contributions are:

1. Demonstrating that DMBRL methods can suffer from a significant performance gap when training from scratch in OpenAI Gym versus DMC environments.

2. Analyzing potential causes for this discrepancy and applying modern mitigation approaches, which ultimately fail to consistently resolve these problems. These results provide insight into the factors affecting performance across different environments.

3. Dramatically accelerating the DMBRL experimentation process with a new JAX-based implementation, which achieves up to a $\sim 40\times$ decrease in wall-clock time. This acceleration lowers the computational barrier for researchers to develop (and comprehensively evaluate) DMBRL algorithms. We have released our code here: https://github.com/CLeARoboticsLab/STFL.

## 2. Background

### 2.1. Reinforcement Learning, Model-Based RL, and Dyna-style Algorithms

Reinforcement learning (RL) models agent-environment interactions as a Markov Decision Process (MDP), defined by the tuple $(\mathcal{S}, \mathcal{A}, p, r, \gamma)$. Here, $\mathcal{S}$ is the state space, $\mathcal{A}$ is the action space, probability distribution $p(s'|s, a)$ represents the transition dynamics, $r(s, a)$ is the reward function, and $\gamma \in [0, 1)$ is the discount factor. The goal of the agent is to learn a policy $\pi(a|s)$ that maximizes the expected cumulative reward.

Learning an effective policy often requires extensive interaction with the environment, which can be prohibitively costly and time-consuming. To address this, model-based RL algorithms aim to learn a model of the environment's dynamics, $p_\theta(s', r|s, a)$, to reduce the need for direct environment interaction. Within this family of algorithms exists the Dyna architecture (Sutton, 1991), which aims to learn a world model $p_\theta$, predict hypothetical transitions under that world model, and then use those imagined experiences to accelerate training. One such method is Model-Based Policy Optimization (MBPO) (Janner et al., 2019), which trains an ensemble of probabilistic neural networks to approximate the environment's dynamics and rewards.

Like most off-policy methods, MBPO maintains a replay buffer of the transitions it has experienced in the environment. MBPO supplements this real world data by generating synthetic on-policy rollouts that branch from states in the replay buffer, thereby augmenting the replay data with "imagined" transitions. This partially synthetic dataset is then used to train both the actor, a neural network that determines the optimal action based on the current state, and the critic, a neural network that estimates the expected return given a choice of action and state. For MBPO this process is done using the model-free algorithm Soft Actor-Critic (SAC) (Haarnoja et al., 2019a), a widely used, reliable method for continuous control tasks. MBPO has demonstrated strong performance on OpenAI Gym benchmarks, often greatly surpassing traditional model-free approaches in sample efficiency. Due to its reputation for strong performance, many recent algorithms are based on MBPO, e.g. (Lai et al., 2020; Li et al., 2024; Dong et al., 2024; Zheng et al., 2023; Wang et al., 2023; Lai et al., 2021).

Another recent extension of the Dyna line of model-based RL is Aligned Latent Models (ALM) (Ghugare et al., 2022). Rather than generating synthetic data in the true state-action space like MBPO, ALM jointly learns observation representations, a world model that predicts next representations given the current representation, and a policy that acts in the representation space. It then uses the Deep Deterministic Policy Gradient (DDPG) algorithm (Lillicrap et al., 2019) for off-policy learning and uses synthetic trajectories to train the latent policy only. Like MBPO, ALM has demonstrated strong performance across OpenAI Gym tasks, while also requiring substantially less wall clock time than MBPO for training, as shown in Figure 2.

Finally, a particularly notable DMBRL approach is DreamerV3 (Hafner et al., 2025), which, unlike the other DMBRL methods presented in this work, was not tested in Gym, but has state-of-the-art sample efficiency in DMC. Like ALM, DreamerV3 optimizes a policy in the latent space using imagination-based trajectories and actor-critic learning. Unlike ALM, which uses a single objective for optimizing both

the policy and latent model, DreamerV3 separates these processes and learns a latent world model using multiple objectives. Please refer to Appendix A for further details.

## 2.2. Benchmarks: OpenAI Gym and DeepMind Control

OpenAI Gym (Brockman et al., 2016) is a widely-used benchmark suite for RL algorithms, providing a variety of environments, including continuous control tasks with and without the MuJoCo physics engine (Todorov et al., 2012) as the physics backend (in this paper, we restrict our attention to Gym environments which use MuJoCo). The DeepMind Control (DMC) Suite (Tassa et al., 2020) provides a larger set of continuous control tasks based on MuJoCo and is designed to provide a more challenging and comprehensive evaluation of control algorithms.

Significant differences exist between DMC and OpenAI Gym in terms of physical parameters, reward structures, and termination conditions. We present an extended discussion of modeling differences and their possible connections to the performance gap we investigate in this paper in Appendix C.

## 3. Performance Gap of Dyna-based MBRL Across Benchmarks

In this section, we put forth the following hypothesis: incorporating Dyna-style modifications into otherwise successful off-policy algorithms can prevent them from improving beyond their performance at initialization — that of a randomly initialized policy. In support of this hypothesis we present empirical evaluations of MBPO's performance across six OpenAI Gym tasks and six challenging robotics tasks (Nikishin et al., 2022) from DMC using our JAX-based (Bradbury et al., 2018) implementation of MBPO. For the OpenAI Gym benchmark, we demonstrate that our implementation matches the performance produced by the original PyTorch (Paszke et al., 2019) MBPO implementation (Janner et al., 2019) and ALM (Ghugare et al., 2022). Furthermore, we show that MBPO dramatically outperforms its "no Dyna" base off-policy algorithm in OpenAI Gym, Soft Actor-Critic (SAC) (Haarnoja et al., 2019a), even when both share identical hyperparameters and architectures.

In contrast, in DMC (Tassa et al., 2020) we observe that MBPO struggles to achieve any policy improvements in six out of fifteen[1] of the challenging environments that we tested on when training from scratch. Because MBPO differs from SAC only due to Dyna-style "enhancements," these results demand a deeper investigation into whether the default SAC hyperparameters, the model ensemble, or other factors induce an otherwise healthy SAC implementation to fail.

---

[1]Training curves for MBPO in all fifteen challenging DMC environments are provided in Appendix G in Figure 11.

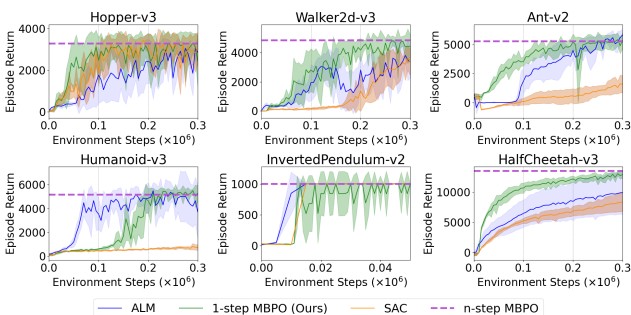

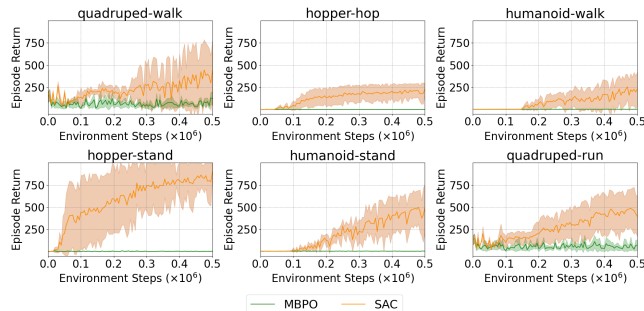

*Figure 3.* 1-step MBPO consistently achieves higher final episodic returns and demonstrates faster convergence compared to SAC. Likewise, 1-step MBPO either matches or surpasses ALM in terms of sample efficiency across all but one task. Solid curves correspond to the mean and shaded regions represent the standard deviation across six trials. The dotted line represents the episodic return of the original $n$-step MBPO at the end of training.

*Figure 4.* Performance of MBPO and SAC on six DMC benchmarks. MBPO exhibits no policy improvement across these environments when Dyna-style enhancements are used, but can immediately make policy improvements without them (i.e. SAC).

This performance gap extends beyond MBPO into other model-based RL algorithms in the Dyna family (Sutton, 1991). To this end, we deploy the recently developed ALM (Ghugare et al., 2022) method across the same six[2] challenging DMC tasks and observe the same phenomenon.

### 3.1. Results in OpenAI Gym

As depicted in Figure 3, when training from scratch our implementation of MBPO using 1-step synthetic rollouts consistently outperforms SAC across six OpenAI Gym environments, and achieves the same sample efficiency as the original $n$-step Pytorch MBPO implementation. These findings agree with those of the original MBPO paper — augmenting policy training with single-step synthetic rollouts produces an algorithm with strong performance (Janner et al., 2019). This affirms both the accuracy of our re-implementation and the sample efficiency gains of 1-step MBPO in OpenAI Gym tasks.

Here, 1-step and $n$-step refer to the length of synthetic rollouts simulated using the learned world model. It is common to prefer shorter rollouts, however, in order to mitigate compounding model errors (Janner et al., 2019; Sikchi et al., 2022); indeed, we demonstrate that these errors are quite substantial in Section 4.1. Combining the results in Figure 2 and Figure 3 we can conclude that our JAX-based implementation of 1-step MBPO matches the original implementation's (Janner, 2019) $n$-step MBPO's performance with much greater wall-clock speed, and as such we will henceforth refer to 1-step MBPO as simply "MBPO."

### 3.2. Results in the DeepMind Control Suite

In contrast, MBPO's performance when training from scratch drops significantly in the DMC environments despite using the exact same hyperparameters and model structure as for the successes in OpenAI Gym. Full results across all fifteen challenging DMC environments are provided in Figure 11 in Appendix G. These results show the complete range of MBPO's performance, including scenarios where MBPO solves tasks more slowly than SAC, matches SAC's sample efficiency, or fails to make any policy improvements.

In this section, we focus on six continuous control tasks where MBPO fails to improve the policy when training from scratch, cf. Figure 4. These results are particularly striking because several of these tasks have high-level analogues in OpenAI Gym when accounting for reward structures, termination conditions, and physics parameter differences.

These results indicate that the recent observation of (Voelcker et al., 2024) — i.e., that MBPO cannot reliably solve the `hopper-hop` environment in DMC — indicates a much broader trend. Moreover, unlike prior work, when we consider these findings alongside the results in Section 3.1, we see that *across multiple environments and two seemingly similar benchmarks, MBPO consistently struggles to improve upon a random policy, let alone train a competent one.* Because one can recover SAC from MBPO by removing the Dyna-style actor/critic updates based on synthetic data, we can conclude that these Dyna-style "enhancements" are the culprit behind MBPO's failure in the DMC environments shown in Figure 4.

### 3.3. Beyond MBPO: Another Dyna-based Algorithm

In this section, we show that the conclusions in Section 3.2 extend beyond MBPO to another member of the Dyna family, Aligned Latent Models (ALM) (Ghugare et al., 2022). We choose ALM for two reasons. First, unlike MBPO,

---

[2]Training curves for ALM in all fifteen challenging DMC environments are provided in Appendix G in Figure 17.

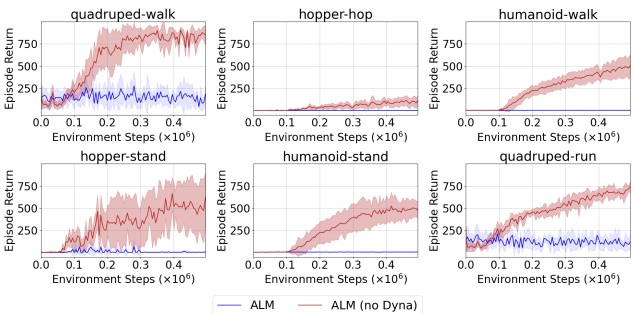

*Figure 5.* Performance of ALM with and without Dyna-style enhancements on six DMC benchmarks. Like MBPO, ALM shows no policy improvement with Dyna-style enhancements, but performs strongly without them (6 trials, $\pm$ std).

ALM is built on top of DDPG, which allows us to test Dyna in a deterministic policy framework, in contrast to SAC's stochastic approach. Second, DDPG's reliance on Gaussian action noising offers a distinct exploration strategy compared to SAC's entropy-based exploration.

To evaluate whether Dyna-based "enhancements" negatively impact ALM, we repeated the same experiment as for MBPO across six[3] challenging DMC tasks. We train ALM with and without Dyna-style enhancements using identical hyperparameters, and compare their performance. This "no-Dyna" variant of ALM is inspired by (Ghugare et al., 2022) where the authors replaced actor training via synthetic model rollouts with a TD3 (Fujimoto et al., 2018) actor loss that relies only on real transitions.

The results of this experiment are shown in Figure 5, and, as with MBPO, the base off-policy algorithm performs quite well, while its Dyna-style counterpart (i.e. ALM) fails to make any policy improvements when training from scratch. This result, when combined with those of Section 3.2, provides clear evidence that *Dyna-style enhancements can prevent improvement beyond the performance of a randomly initialized policy across many environments*.

Throughout the remainder of this paper, we analyze the underlying causes of these performance discrepancies through the lens of MBPO, and explore potential solutions. By doing so, we aim to identify whether these challenges are intrinsic limitations of Dyna-style algorithms in certain settings or if they can be mitigated in a general manner.

[3]Training curves for ALM in all fifteen challenging DMC environments are provided in Appendix G in Figure 17.

**Key insights:**

- MBPO consistently matches or outperforms the sample efficiency and final episodic return of its "no Dyna" variant (i.e. SAC), across *OpenAI Gym* environments.

- MBPO consistently *under*performs the sample efficiency and final episodic return of its "no Dyna" variant (i.e. SAC), across *DMC* environments.

- This same performance gap can be observed for ALM, another Dyna-based RL algorithm, suggesting these issues may be endemic to a larger subclass of the Dyna algorithm family.

## 4. The Effect of the Predictive Model on Algorithm Performance

### 4.1. Is High Model Error the Problem?

In Section 3, we saw that Dyna-style "enhancements" caused both MBPO and ALM to dramatically underperform their "no Dyna" counterparts in DMC, whereas the same enhancements led to performance gains in OpenAI Gym. Since the predictive model ensemble is central to MBPO's sample efficiency gains over model-free methods, we focus on this first as a source of MBPO's issues. Studies have shown that high model error causes significant performance degradation (Gu et al., 2016; Rajeswaran et al., 2017), and models are most effective over short to moderate planning horizons (Sikchi et al., 2022; Holland et al., 2019) if compounding model errors can be mitigated (Talvitie, 2014; Buckman et al., 2018). We find that severe model error in many DMC environments makes even 1-step rollouts unrealistic, rendering multi-step methods unproductive.

To quantify this effect, we introduce the percent model error, defined as:

$$\frac{\|\hat{y} - y\|_2}{\|y\|_2} \times 100,$$

where $\hat{y}$ denotes the model's predicted next observation and reward, and $y$ is the corresponding ground-truth target. This normalized metric allows for fair comparison across tasks with differing scales.

The relationship between percent model error and performance degradation in MBPO is challenging to assess, particularly within DMC environments. We investigated this via comparisons of percent model errors across the training distribution (i.e., the replay buffer) for six tasks in both OpenAI Gym and DMC (cf. Figures 12 and 13 in Appendix G) instead of relying on computationally expensive on-policy rollouts. We observed that in all six OpenAI Gym tasks the model error converges significantly below 25%. In con-

trast, for the six DMC tasks where MBPO fails, the tasks fall into two categories. The `hopper` tasks exhibit model errors that converge above $100\%$, while the remaining tasks have model errors converging above $25\%$ as training progresses. Notably, all of these environments fail to improve policy performance beyond that of a random policy (recall Figure 4).

While modeling errors in MBPO for DMC environments are not universally larger than those in OpenAI Gym, their impact is difficult to contextualize without further analysis. To investigate this impact, and inspired by previous work (Kalweit & Boedecker, 2017), we vary the synthetic-to-real data ratio, $S$, which represents the proportion of synthetic rollouts in each training batch. By varying $S$, we can evaluate a spectrum between MBPO ($S \to 0.95$) and SAC ($S \to 0$), thereby systematically studying the effect of synthetic data on policy performance and assessing the significance of modeling errors in degrading MBPO's performance (Gu et al., 2016).

Focusing on `hopper-stand` and `humanoid-stand`, which represent the two categories of model error magnitude and highlight where MBPO consistently fails with a learned model, we find a clear trend: increasing $S$ reduces episodic returns (cf. Figure 14 in Appendix G). This effect is particularly evident for `humanoid-stand`, where even minimal synthetic data severely degrades performance, highlighting the detrimental effect of model errors on MBPO.

To rule out tuning issues, we performed comprehensive hyperparameter sweeps for `hopper-stand` and `humanoid-stand` (cf. Appendix F), since introducing any synthetic data can induce failures in MBPO with the default hyperparameters. We swept over the key hyperparameters related to the training and utilization of the predictive model, including the size of the model's hidden layers, the interval at which the model is retrained, the learning rate, and the number of training steps. None provided any improvement to policy performance across training despite reductions in model-error, which suggests that deeper architectural and algorithmic changes are required. Reduction of model bias remains an open issue in the broader MBRL literature (Wang et al., 2019; Deisenroth & Rasmussen, 2011; Luo et al., 2024) and we aim to address this in future work as part of a broader solution to Dyna-style algorithm issues identified here.

### 4.2. What If We Had a Perfect Model?

These results raise a fundamental question: if we had access to a perfect model of the environment, would MBPO outperform SAC in DMC, as it does in OpenAI Gym? Addressing this question allows us to bypass the long-standing challenge of training a reliable predictive model (Atkeson & Schaal, 1997), and directly evaluate whether MBPO with

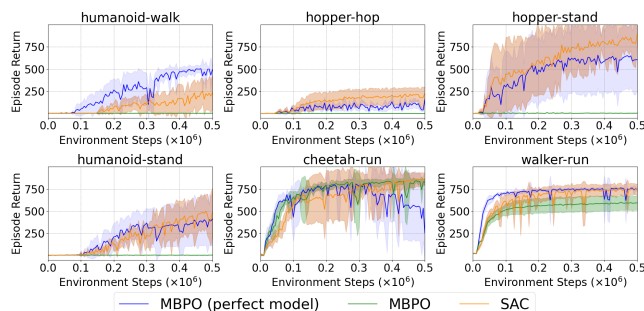

*Figure 6.* Comparing MBPO with a perfect predictive model to SAC and MBPO with a learned model. Solid curves correspond to the mean and shaded regions represent the standard deviation across six trials.

the default hyperparameters and access to perfect rollouts can improve upon its base off-policy method (i.e., SAC) across multiple environments in DMC.

To answer this question, we modified the reset procedure of the DMC simulator to process arbitrary initial states provided by the user. Using this augmented simulator, we generated real trajectories starting from states sampled randomly from the replay buffer during training. All other hyperparameters for MBPO were left unchanged. The results for six DMC environments are presented in Figure 6.

Access to a perfect predictive model produces compelling yet unsatisfying results. In particular, across these six environments — which include a mix of cases settings in which MBPO previously had either promising results or no policy improvements — *even with a perfect model, MBPO cannot outperform its base off-policy method consistently.*

In OpenAI Gym, even MBPO with a learned model achieves sample efficiency comparable to or exceeding SAC across all environments (cf. Figure 3). However, in DMC, while a perfect model enables policy improvement in environments where MBPO with a learned ensemble fails completely, it does not consistently surpass SAC's sample efficiency in four out of the six environments shown in Figure 6.

It becomes clear that even with an idealized model, modeling errors alone cannot fully explain the failings of MBPO in DMC. The `hopper` tasks exhibit significantly larger model errors compared to `humanoid-stand` and other tasks (Figure 13), yet even with the perfect model, MBPO does not achieve the same level of success for `hopper` (Figure 6) as SAC. This observation indicates that further analysis is needed to investigate other sources of these shortcomings.

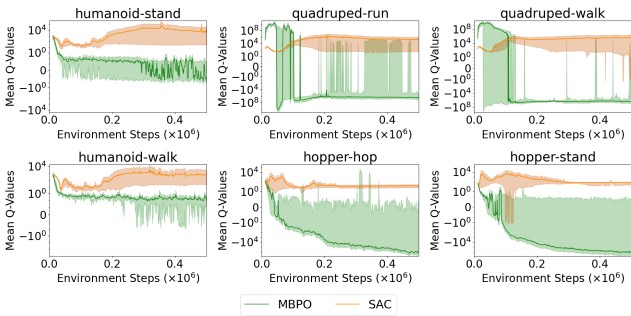

*Figure 7.* Mean Q-values (6 seeds) with shaded regions showing min-max range. MBPO's critic either massively underestimates or predicts no return when compared to SAC. Overestimation is magnified in `quadruped` tasks with Dyna-style enhancements. Results across all 15 DMC tasks are provided in Figure 16.

### 4.3. Investigating Underlying Failure Mechanisms Induced By Model Bias

Since even access to the true environment dynamics fails to resolve MBPO's poor performance in DMC, we explored additional possible failure modes beyond model bias that could hinder MBPO's success. These additional failure modes relate to the concept of *replay ratio* (RR).

In off-policy reinforcement learning, the replay ratio refers to the frequency of RL training updates relative to the number of environment interactions. For example, SAC performs one update per environment interaction (RR=1), while MBPO performs twenty updates (RR=20), resulting in a much higher replay ratio. MBPO's high replay ratio, combined with its lack of performance improvement beyond the randomly initialized policy, suggests the presence of critic divergence — a well-known challenge in off-policy RL (Thrun & Schwartz, 1993) which is often exacerbated by overly-replaying stale data during training. Recent studies in high replay ratio off-policy RL (Nauman et al., 2024; Hussing et al., 2024) have highlighted two forms of critic divergence: overestimation and underestimation.

Overestimation, common in offline and online off-policy methods with high replay ratios, arises when a learned Q-function is queried for state-action pairs that are out-of-distribution for the training data (Thrun & Schwartz, 1993). Online methods typically mitigate this by using real experiences to explore high-reward regions, naturally correcting overestimation through interaction. Conversely, underestimation occurs in unseen regions where the Q-function is overly pessimistic. Unlike overestimation, underestimation is harder to address, as it requires purely exploratory actions to visit these regions, which policies inherently avoid due to their focus on maximizing expected return.

To investigate critic divergence, we measured the average

critic Q-values throughout training for SAC and MBPO in DMC environments where MBPO fails, and report the results in Figure 7. MBPO's Q-values exhibited significantly greater divergence despite identical hyperparameters. This suggests that inaccurate synthetic transitions drive the Q-value estimation issues and, by extension, hinder learning.

Since the MBPO model predicts both next states and rewards inaccurately even on its training distribution, and because synthetic data dominates the training set by default in MBPO, the actor and critic repeatedly encounter infeasible transitions and unrealistic returns. We hypothesize that these synthetic transitions conflict with actual replay data, causing severe critic target non-stationarity and substantial Q-value estimation errors.

To validate this hypothesis, we compared the Q-values obtained using both a learned model and a perfect model throughout training.[4] With a perfect model, massive underestimation is much less pronounced, implicating modeling errors as the primary driver of this issue. However, per the discussion of Section 4.2, even without massive underestimation MBPO fails to match SAC's performance.

Since critic divergence stems from learned functions extrapolating in an unconstrained manner, we employed Layer Normalization (Ba et al., 2016a) as a regularization technique as in (Ball et al., 2023) to bound the critic output. This technique has proven quite successful in previous work (Nauman et al., 2024), but even after partly mitigating critic divergence, MBPO still cannot outperform SAC consistently as shown in Appendix G in Figure 20.

> **Key Insights:**
>
> - MBPO's learned model struggles to make accurate predictions in DMC, even on the training distribution.
>
> - A perfect 1-step model improves MBPO in DMC but still fails to match or outperform SAC, as was previously achieved in Gym.
>
> - Predictive model errors and non-stationary critic targets induce critic divergence.
>
> - Layer Normalization mitigates critic divergence, but does not allow MBPO to succeed.

## 5. If It's Not Just The Model Could It Be The Learning Dynamics?

Section 4 suggests that errors in the predictive model contribute to MBPO's poor performance in DMC, but cannot fully explain the issues identified in Section 3. This obser-

---

[4]These results are in Appendix G in Figure 18.

vation leads us to investigate how learning dynamics and network plasticity in MBPO might impact the predictive model or the base off-policy method's abilities.

Plasticity loss — where prolonged training diminishes a network's capacity to learn new tasks — has been studied in off-policy RL (Lyle et al., 2022) and model-based RL (Qiao et al., 2023), with successful applications in DMC (Nikishin et al., 2022; D'Oro et al., 2023). High replay ratios in off-policy RL are known to exacerbate plasticity issues, as both the predictive model and the learned Q-function face continually changing data distributions and using a high replay ratio forces them to learn to solve a sequence of similar, but distinct, tasks (Dabney et al., 2021).

Qiao et al. (2023) demonstrate that periodic reinitialization of the learned model parameters can mitigate the loss of plasticity in model-based RL and enhance model accuracy. Therefore, to determine if a loss of plasticity is contributing to MBPO's failures, we completely reset all parameters of the predictive model every $2 \times 10^4$ environment steps, which is a frequency aligned with recommendations from previous work (Qiao et al., 2023; D'Oro et al., 2023) for a replay ratio of 128. We also experimented with both more conservative intervals (e.g., every $1.28 \times 10^5$ environment steps) and intermediate values, but found that results were indistinguishable from those in Figure 4 regardless of reset interval. These findings suggest that model plasticity is not the primary cause of low model accuracy in MBPO.[5]

Next, we investigated whether plasticity issues were affecting the networks that comprise the off-policy base of MBPO — the actor, critic, target critic, and the automatically tuned temperature in SAC. Unlike the investigation of the model losses due to plasticity, we reset all of the aforementioned networks, except the predictive model, every $2 \times 10^4$ interactions. In Figure 8, we see that in both `quadruped` tasks performing periodic resets allows MBPO to not only improve its policy from the initial random policy, but outperform SAC by a large margin. Additionally we see that in the `humanoid-walk` task performing periodic resets allows MBPO to improve from the initial random policy, but it still underperforms SAC. In all other tasks MBPO with periodic resets still cannot improve its policy beyond the initial random policy. We can conclude that plasticity issues related to these base off-policy components do, at least partially, account for some failings of MBPO.

Nevertheless, alleviating plasticity loss is not a universal solution. Of even more significance, the periodic resets applied in MBPO are equally applicable to SAC. As shown in Figure 8, SAC with periodic resets *significantly* outperforms MBPO with the same resets on most tasks. This highlights

---

[5]Training curves for model resets every $2 \times 10^4$ environment steps may be found in Figure 20 in Appendix G.

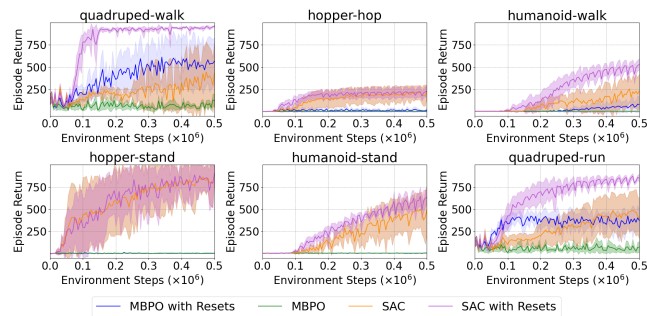

*Figure 8.* Comparison of MBPO and SAC with and without periodic resets of the actor, critic, target critic, and automatically tuned temperature. While periodic resets help MBPO improve policy performance beyond the random initial policy in some environments, SAC with resets consistently outperforms MBPO across all tasks. Full results are in Appendix G and Figure 15.

that Dyna-style enhancements can degrade the performance of their off-policy base algorithm in DMC.

> **Key Insights:**
>
> - Loss of plasticity in the predictive model does not contribute to MBPO's failures.
>
> - Loss of plasticity in the off-policy base (actor, critic, target critic, and temperature) is only partially responsible for MBPO's limitations.
>
> - MBPO's Dyna-style enhancements exacerbate plasticity loss in the off-policy base, leading to worse performance compared to SAC.
>
> - SAC continues to significantly outperform MBPO in DMC when periodic resets are applied to both algorithms.

## 6. Conclusion

In this work, we have illustrated a surprising and consistent performance gap of Dyna-style model-based reinforcement learning algorithms when applied across diverse benchmark environments. While these algorithms can excel in OpenAI Gym benchmark tasks, their performance degrades significantly in the DeepMind Control Suite (DMC), even though tasks in these benchmarks share broadly similar characteristics and the same underlying physics engine. Our extensive experiments, reinforced by modern techniques designed to bolster both model-free and model-based methods, reveal that synthetic rollouts — central to Dyna's gains in OpenAI Gym — can arrest policy improvement rather than enhance it when deployed across more diverse environments. Concretely, for all 15 DMC tasks we examined, adding model-generated samples consistently undermined

both sample efficiency and wall-clock performance relative to simpler, model-free off-policy algorithms with identical hyperparameters. In short, there is no free lunch.

These findings suggest that our community's reliance on a limited set of benchmarks may have contributed to an inflated view of the generality and robustness of Dyna-style approaches. Just as importantly, our results highlight systemic issues within the research community and publication ecosystem: there are few incentives — monetary or otherwise — for conferences, journals, and researchers to invest in the critical re-examination of widely cited methods. Model-Based Policy Optimization (MBPO), in particular, is an instructive example. Although it is frequently reproduced, highly cited, and even featured in a reproducibility study accepted at a major conference (Liu et al., 2020), our analysis shows that its touted advantages fail to carry over to an equally conventional testbed (DMC). This underscores the need for a cultural shift towards more rigorous, critical, and comprehensive evaluation practices, including active self-policing of influential algorithms and promoting rigorous follow-up studies that challenge established claims.

Looking ahead, we have not solved the pervasive issues underlying the subclass of Dyna-style methods investigated in this work, but we have made progress towards facilitating such work by identifying that there is a problem, investigating potential causes and solutions, and providing code that significantly speeds up evaluation procedures. We hope that this work provides a building block from which our community can dissect, diagnose, and ultimately address the shortcomings that currently limit Dyna's utility.

Recognizing that our work is not exhaustive, we also have included a frequently asked questions section in Appendix A to address common questions, provide context, and foster collaboration. We intend to update it continually, incorporating insights and feedback from the broader research community as we refine our methods.

## Acknowledgements

This work was supported by the National Science Foundation under Grant No. 2409535. We would like to thank Harshit Sikchi, Eugene Vinitsky, Siddhant Agarwal, and Max Rudolph for insightful discussions and their feedback on this paper.

## Impact Statement

This paper underscores the importance of robust and diverse benchmarking in the Machine Learning community to ensure that reported progress translates to meaningful advances. By pointing to the limitations of Dyna-style methods in broader contexts, we encourage the community to critically re-evaluate widely adopted approaches and the systems that reward them. Addressing systemic issues, such as the incentives for replicating influential results without deeper scrutiny, is essential to fostering meaningful progress in the field.

Furthermore, our work highlights the barriers to accessibility in reproducibility and evaluation. Many influential methods, including those analyzed in this study, remain difficult to reproduce due to the significant computational resources required to replicate results or run new experiments. Such challenges prevent broader participation in the refinement and critical evaluation of algorithms, especially from researchers with limited access to cutting-edge hardware. To help address these concerns for Dyna-style methods, we provide efficient, open-source code that significantly reduces evaluation costs and facilitates more inclusive collaboration within the research community.

From an ethical standpoint, this work does not present immediate societal risks, but rather focuses on improving the methodological rigor within reinforcement learning research. By advocating for more comprehensive evaluation practices, reducing resource requirements, and fostering transparency, we hope to encourage accountability and accessibility in research practices.

We believe this work holds value for the community by drawing attention to challenges in generalization, overfitting to benchmarks, accessibility, and the broader need for a culture that prioritizes the evaluation of actual progress. While our study is not exhaustive, we provide a framework for further investigation and invite collaboration to refine and build upon our findings. By enabling the community to address these shortcomings, we aim to facilitate research that leads to more robust, impactful, and accessible algorithms.

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

# A. Frequently Asked Questions

**Are you saying that no Dyna-style model-based RL algorithm can succeed in DMC?**
No. As mentioned at the end of Section 2.1, DreamerV3 (Hafner et al., 2025) achieves state-of-the-art performance in DMC environments. This evidence suggests that there are two subclasses of DMBRL algorithms: those that succeed in DMC and those that do not. This paper highlights two algorithms in the latter subclass, with the aim of inspiring future work to identify algorithmic improvements that enable all DMBRL algorithms to perform successfully not only in OpenAI Gym or DMC, but across a broader range of tasks.

**Isn't the poor performance of the models evidence that model learning, not Dyna, is the real problem?**
No. Even if we ignore the results of Section 4.2, the difficulty of learning an accurate model is central to the viability of Dyna-style methods (Sutton, 1991). While the learning of the model itself may lie outside the scope of methods based on the Dyna architecture, these algorithms fundamentally rely on the model to generate useful synthetic experience. If the model struggles, so does the entire method. The success of Dyna-style approaches, especially MBPO and ALM, hinges on their ability to make progress even with imperfect models, making model learning an inseparable and integral part of their evaluation.

Notably, the model learning architectures proposed alongside MBPO and ALM have demonstrated strong performance in OpenAI Gym environments (cf. Section 3.1), yet they struggle in DeepMind Control (DMC) (cf. Section 3.2 and Section 3.3). Given the similarities between these environments (cf. Appendix C), this discrepancy highlights the need to further investigate where and why these models falter in DMC, how to improve their robustness, and how to improve the underlying Dyna algorithms' ability tolerate inevitable model errors.

**Did you consider overfitting of the model as a potential source of error?**
Yes. We did not find any significant overfitting of the model in either the OpenAI Gym environments or in DMC. Further, we performed intermittent model resets in Section 5 as a means to mitigate this issue if overfitting was occurring throughout training (Qiao et al., 2023). As discussed in Section 5, resets do not resolve the performance gap.

**Did you consider running <insert experiment here>?**
This paper does not (and cannot) exhaustively consider all potential of sources of the performance gap we observed. If you can think of additional experiments to try, we would love to hear from you.

# B. Timing Claim for MBPO Compared to Our Implementation

Based on previous work (Ghugare et al., 2022), MBPO's average time per environment step was approximately 0.6 seconds across Gym environments. Even if we optimistically reduce this to 0.5 seconds per step (to account for our usage of slightly more modern computational hardware) the time required to run 6 seeds across the 6 DMC environments shown in Figure 4 for 500k environment steps is: $\frac{0.5 \times 500000 \times 6 \times 6}{(3600 \times 24)} = 104$ days.

In contrast, our code was directly timed during experimentation. The total runtime for six seeds across six environments and 500k steps came out to approximately 4 days on an NVIDIA RTX A5000 GPU.

# C. Extended Discussion on Benchmark Modeling Differences

As discussed in Section 2.2, OpenAI Gym and DM Control (DMC) use different reward structures, physical parameters, and termination conditions. These differences certainly could contribute to Dyna-style algorithms failing in DMC but not Gym. However, both are equally conventional and widely accepted testbeds for reinforcement learning. This is evidenced by the fact that DMC has been successfully used for demonstrating RL with not only proprioceptive observations (D'Oro et al., 2023; Nikishin et al., 2022), but for visual observations too (Laskin et al., 2020; Hafner et al., 2020; Laskin et al., 2020; Yarats et al., 2022).

It is important to emphasize that the goal of this work is not to pinpoint specific environmental factors that might cause DMBRL to fail. Instead, we aim to demonstrate that while DMBRL performs well on one benchmark, it largely fails on another that, at first glance, appears quite similar. This observation raises concerns about the robustness of DMBRL algorithms. Rather than delving into the algorithmic causes of this issue as we have for the majority of this paper, this section highlights and discusses key differences between the two benchmarks, providing a foundation for future work to

investigate environment-specific failures of DMBRL algorithms.

Gym environments typically feature unnormalized rewards and explicit termination conditions, while DMC provides normalized rewards and lacks termination conditions. Despite these differences, our discussion below shows that reward and termination conditions alone do not fully explain performance disparities. Given the substantial similarity between many tasks across the two frameworks, we can compare their MDP specifications side-by-side, focusing on termination conditions first.

In a unique experiment in the `hopper` environment, previous work (Voelcker et al., 2024) removed only termination conditions from Gym and ran MBPO, ALM, and SAC. While removing termination conditions caused MBPO to fail, ALM was still able to train a successful policy. However, as we show in Section 2.2, as soon as MBPO or ALM are deployed in DMC, which also has no termination conditions, ALM fails. This evidence shows that at least for environments that are broadly similar to `hopper` termination alone does not explain DMBRL failing in DMC.

Conversely, MBPO performed relatively well on `walker` tasks in both Gym and DMC environments, as shown in Figures 3 and 6, despite DMC lacking termination conditions. Interestingly, ALM exhibited a different pattern, failing on `walker` in DMC but succeeding in Gym. Additionally, switching frameworks reversed the relative performance of SAC and MBPO for `walker`. In the `humanoid` environment, both MBPO and ALM succeeded in Gym but failed entirely in DMC. These results indicate that termination conditions alone do not account for DMBRL method failures across multiple environments that are broadly similar, as performance varies across frameworks and tasks.

The physical differences in robot models between the two frameworks are also notable, particularly in the configuration of joints, actuators, and contact dynamics. For instance, at a high level, OpenAI Gym's `humanoid` is built to be more stable and controlled, using higher stiffness and damping to keep its movements steady and balanced. On the other hand, DMC's `humanoid` is designed for smoother and more natural motion, with lower stiffness and damping in peripheral joints (i.e. the arms, legs), but still adding extra stiffness where needed for stability (i.e. in the hips and torso). The Gym `humanoid` also has armature (rotational inertia added by actuators) values that vary by joint, which helps it move quickly and adjust better to different tasks. DMC's `humanoid` keeps the same armature value everywhere, making it simpler but less flexible for specific challenges.

Similar trends are observed in the `hopper` environment. Gym's model emphasizes precision and stability with higher damping and armature values, while DMC's design favors fluidity with lower damping and moderate, uniform armature values. However, this trend does not extend to the `walker` model, where both frameworks use identical damping and armature values.

These differences may very well contribute to the performance gap identified in Section 3. However, the analysis of Section 4 — wherein we find that MBPO's learned world model tends to exhibit substantially higher error in DMC than in Gym, but that even with access to a perfect model MBPO cannot reliably outperform SAC — indicates that this is only part of the story.

There are numerous other differences across the various environments in Gym and DMC that we cannot enumerate here, but one final distinction lies in the numerical integration schemes used by the two frameworks. Across the environments we have discussed in this section, Gym employs a smaller timestep and the 4th-order Runge-Kutta method for numerical integration, while DMC uses MuJoCo's default semi-implicit Euler method (Tassa et al., 2020). These differences result in Gym having a shorter effective horizon for tasks that take identical numbers of environment steps and higher accuracy in simulating dynamics. Such variations could induce significant discrepancies even in environments that are otherwise identical in their physical specifications.

In summary, while termination conditions, physical parameters, and numerical integration schemes differ between OpenAI Gym and DM Control, no single factor fully explains the performance variations observed in DMBRL methods. Instead, the interplay between these elements, and likely ones we have not discussed, contributes to the disparities observed when deploying DMBRL between different benchmarks.

If you have other ideas about how the differences between the Gym and DMC benchmarks may be contributing to the performance gap we identify in this work, we would love to hear about them. Please reach out to the corresponding author.

# D. Implementation Details

## D.1. ALM

The code to run ALM was used with only minimal modifications introduced to allow interfacing with DMC environments. No hyperparameters were changed in either our Gym experiments or in our DMC experiments from what was provided in the original ALM repository. When applying ALM without Dyna-style enhancements we simply replaced the `lambda svg loss` with the `td loss` that is provided in the experimental branch of the ALM repository and kept all hyperparameters fixed. **The specific implementation details below are reproduced with minor modification from (Ghugare et al., 2022)**:

We implement ALM using DDPG (Lillicrap et al., 2019) as the base algorithm. Following prior SVG methods (Amos et al., 2021), we parameterize the encoder, model, policy, reward, classifier and Q-function as 2-layer neural networks, all with 512 hidden units except the model which has 1024 hidden units. The model and the encoder output a multivariate Gaussian distribution over the latent-space with diagonal covariance. Like prior work (Hansen et al., 2022; Yarats et al., 2022), we apply layer normalization (Ba et al., 2016a) to the value function and rewards. Similar to prior work (Schulman et al., 2015; Hafner et al., 2020), we reduce variance of the policy objective by computing an exponentially-weighted average of the objective for rollouts of length 1 to an integer $K$ chosen as a hyperparameter. To train the policy, reward, classifier and Q-function we use the representation sampled from the target encoder. For exploration, we use added normal noise with a linear schedule for the standard deviation (Yarats et al., 2022). All hyperparameters are listed in Table 3.

For additional details see (Ghugare et al., 2022).

## D.2. MBPO

The code for our MBPO implementation is available at https://github.com/CLeARoboticsLab/STFL.

# E. Nominal Hyperparameters for SAC, MBPO, and ALM

*Table 1.* Hyperparameters used for SAC and the SAC component of MBPO.

| Hyperparameters | Value |
|---|---|
| Discount ($\gamma$) | 0.99 |
| Warmup steps | 10000 |
| Minibatch size | 256 |
| Optimizer | Adam |
| Learning rate ($\alpha$) | 0.0003 |
| Optimizer $\beta_1$ | 0.9 |
| Optimizer $\beta_2$ | 0.999 |
| Optimizer $\epsilon$ | 0.00015 |
| Networks activation | ReLU |
| Number of hidden layers | 2 |
| Hidden units per layer | 256 |
| Initial temperature ($\alpha_0$) | 1 |
| Replay buffer size | $10^6$ |
| Updates per step | 20 |
| Target network update period | 1 |
| Soft update rate ($\tau$) | 0.995 |

*Table 2.* Hyperparameters used for training and deployment of Dyna-style enhancements in MBPO.

| Hyperparameters | Value |
| --- | --- |
| Ensemble retrain interval | 250 |
| Minibatch size | 256 |
| Optimizer | Adam |
| Ensemble learning rate | 0.0003 |
| Optimizer $\beta_1$ | 0.9 |
| Optimizer $\beta_2$ | 0.999 |
| Optimizer $\epsilon$ | 0.00015 |
| Networks activation | Swish |
| Synthetic ratio | 0.95 |
| Model rollouts per environment step | 400 |
| Number of ensemble layers | 2 |
| Hidden units per layer | 200 |
| Number of elite models | 5 |
| Number of models in ensemble | 7 |
| Model horizon | 1 |

*Table 3.* Hyperparameters used for ALM with and without Dyna-style enhancements and reproduced from (Ghugare et al., 2022).

| Hyperparameters | Value |
| --- | --- |
| Discount ($\gamma$) | 0.99 |
| Warmup steps | 5000 |
| Soft update rate ($\tau$) | 0.005 |
| Weighted target parameter ($\lambda$) | 0.95 |
| Replay buffer | $10^6$ for humanoid |
| | $10^5$ otherwise |
| Batch size | 512 |
| Learning rate | 1e-4 |
| Max grad norm | 100.0 |
| Latent dimension | 50 |
| Coefficient of classifier rewards | 0.1 |
| Exploration stddev. clip | 0.3 |
| Exploration stddev. schedule | linear(1.0 , 0.1, 100000) |

## F. Hyperparameter Sweeps

We conducted a large-scale hyperparameter sweep to evaluate MBPO's sensitivity in `hopper-stand`. This sweep comprised 93 hyperparameter configurations, with each configuration evaluated across three seeds to capture variation due to initialization and training dynamics, resulting in a total of 279 runs. The full range of values is listed in Table 4. All runs used a synthetic-to-real data ratio of $0.95$ and were trained online for 50k environment steps. No combination of hyperparameters in this sweep yielded consistent performance.

To test whether the findings central to our claims generalized across environments and longer training durations, we conducted an additional sweep using single-seed runs on both `hopper-stand` and `humanoid-stand`. For each environment, we evaluated 81 hyperparameter configurations for 200k environment steps (see Table 5). These experiments targeted learning dynamics across a higher number of environment interactions and helped ensure that the lack of improvement was not simply due to undertraining or insufficient data. No combination of hyperparameters yielded consistent performance.

Since none of the 174 hyperparameter variations across 2 environments and 441 runs led to meaningful performance gains, we omit per-parameter plots for brevity. Instead, Figure 9 and Figure 10 show unlabeled return curves for each configuration. Despite substantial hyperparameter sweeps, return performance remained poor, reinforcing our claim that model-based policy optimization fails to benefit from tuning in this setting.

*Table 4.* Hyperparameter sweep ranges for MBPO on `hopper-stand`.

| Hyperparameter | Values |
|---|---|
| `ensemble_hidden` | {50, 100, 200, 400} |
| `ensemble_lr` | {1.5e-4, 3e-4, 6e-4} |
| `ensemble_interval` | {125, 250, 500} |
| `model_train_steps` | {1250, 2500, 5000, 10000} |
| `model_rollouts_per_step` | {100, 200, 400, 800, 1600} |
| `num_elites` | {1, 3, 5} |

*Table 5.* Hyperparameter sweep ranges used in single-seed runs on both `hopper-stand` and `humanoid-stand`.

| Hyperparameter | Values |
|---|---|
| `ensemble_hidden` | {200, 400, 800} |
| `ensemble_lr` | {1.5e-4, 3e-4, 6e-4} |
| `ensemble_interval` | {125, 250, 500} |
| `model_rollouts_per_step` | {200, 400, 800} |

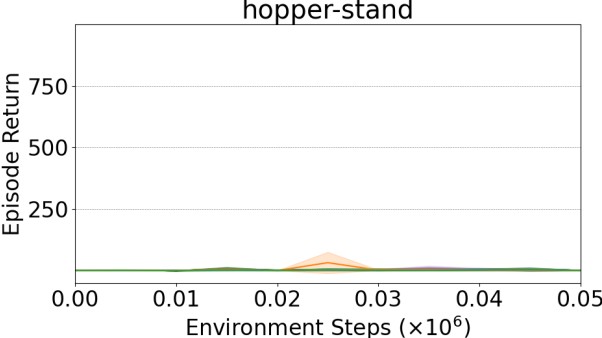

*Figure 9.* Hyperparameter sweep for MBPO trained on the `hopper-stand` environment across 93 hyperparameter configurations (3 trials each, ± std). As none of these configurations yielded meaningful or consistent performance, we omit per-parameter plots for brevity. Despite one seed of one configuration temporarily reaching a return of 93, the policy immediately collapses to near-random performance after further training. For sake of comparison, after 50k environment steps a SAC policy in this same task has an average return of 300.

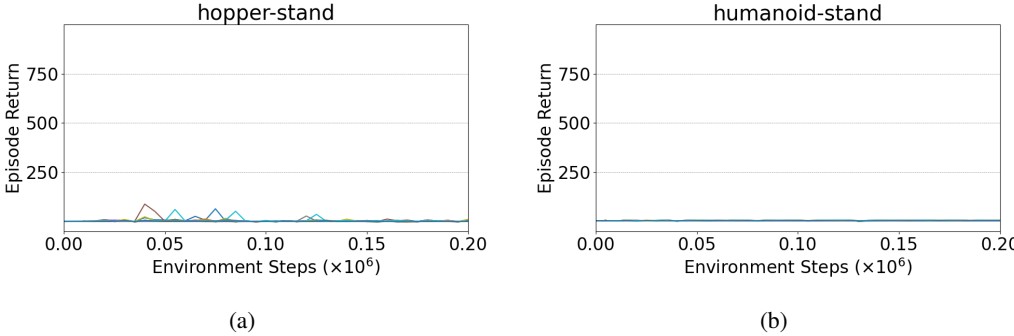

(a)                                                    (b)

*Figure 10.* Hyperparameter sweep for MBPO trained on the `hopper-stand` and `humanoid-stand` environment across 81 hyperparameter configurations each (1 trials each).

# G. Full Experimental Results

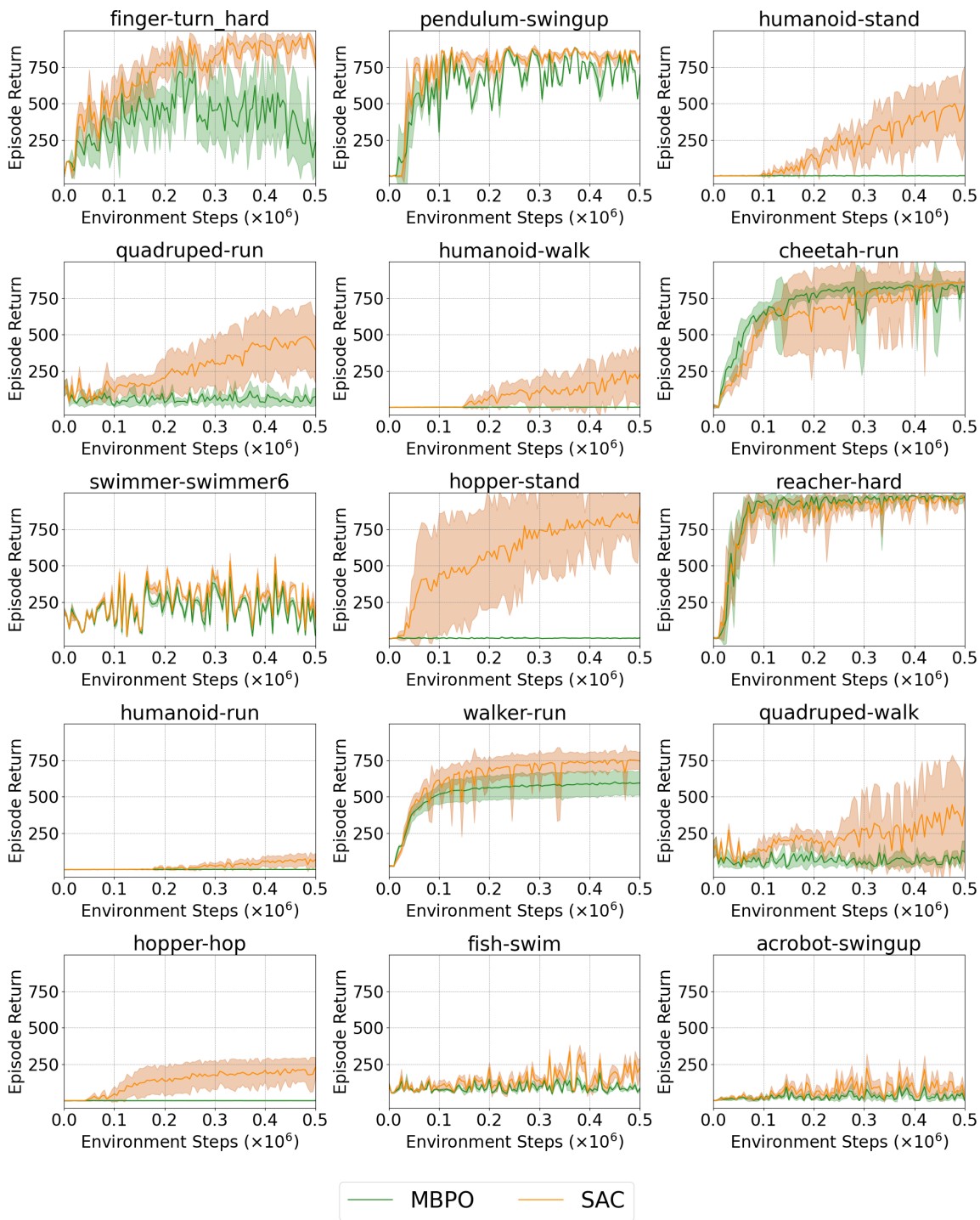

*Figure 11.* Full performance of MBPO and SAC for 15 challenging DMC benchmark tasks as referenced in Section 3.2.

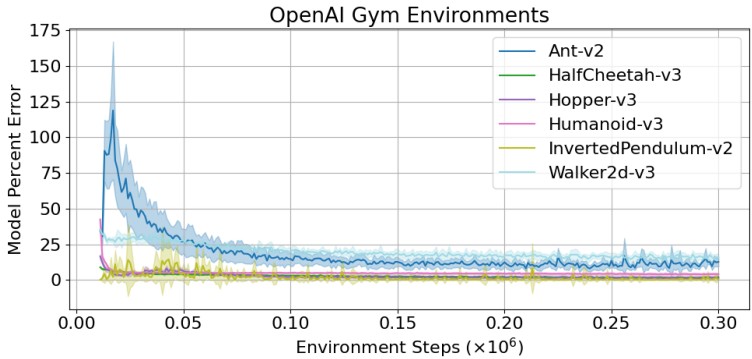

*Figure 12.* Illustration of the change in percent model error during RL training for OpenAI Gym, as measured on the training distribution. (6 trials, ± std).

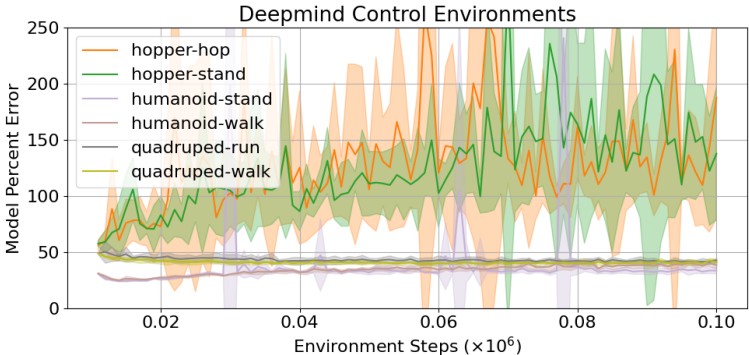

*Figure 13.* Illustration of the change in percent model error during RL training for DMC, as measured on the training distribution. (6 trials, ± std).

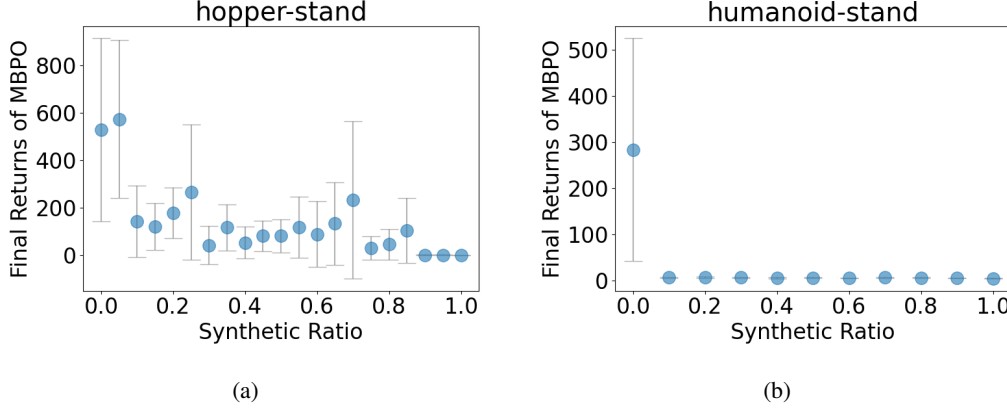

*Figure 14.* Illustration of the relationship between final episodic return and synthetic-to-real ratio used for MBPO training for (a) `hopper-stand` for 6 seeds and 100k steps each, and (b) `humanoid-stand` for 3 seeds and 400k steps each. These results show that more synthetic data in a training batch leads to worse final policy performance for MBPO.

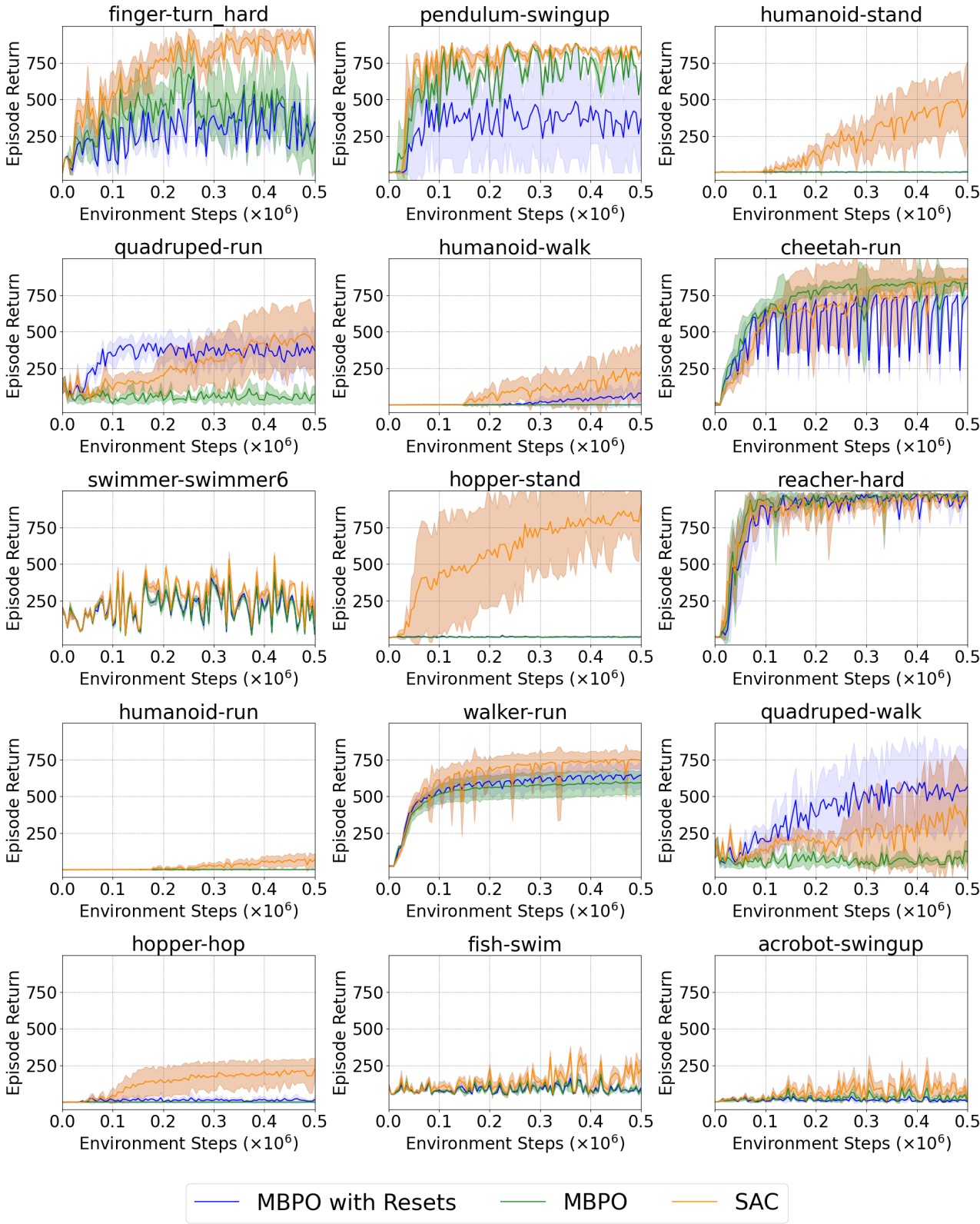

*Figure 15.* Comparing full performance of MBPO, SAC, and MBPO with actor and critic resets for 15 challenging DMC benchmark tasks as referenced in Section 5.

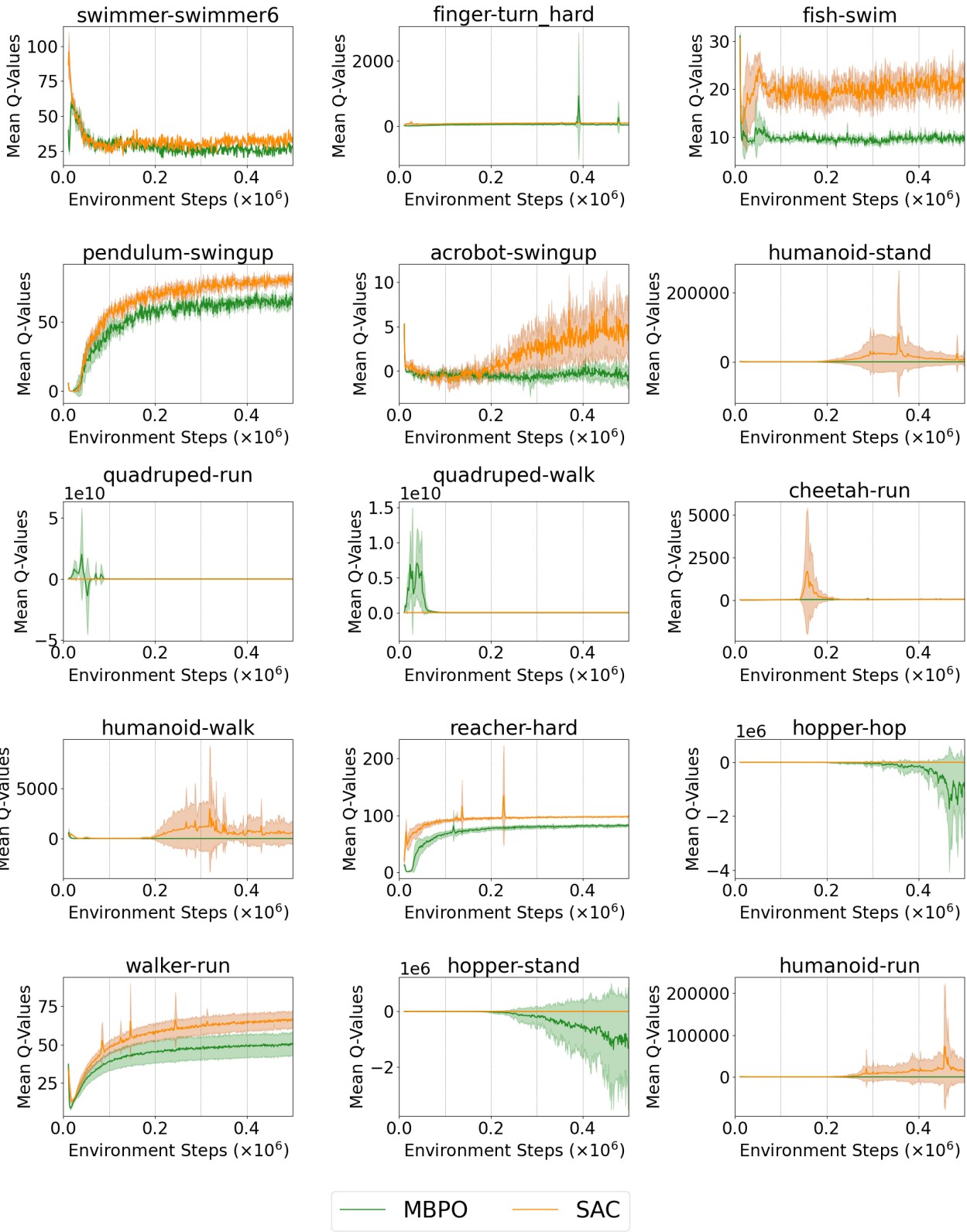

*Figure 16.* Mean Q-values for 15 challenging DMC benchmark tasks as referenced in Section 4.3.

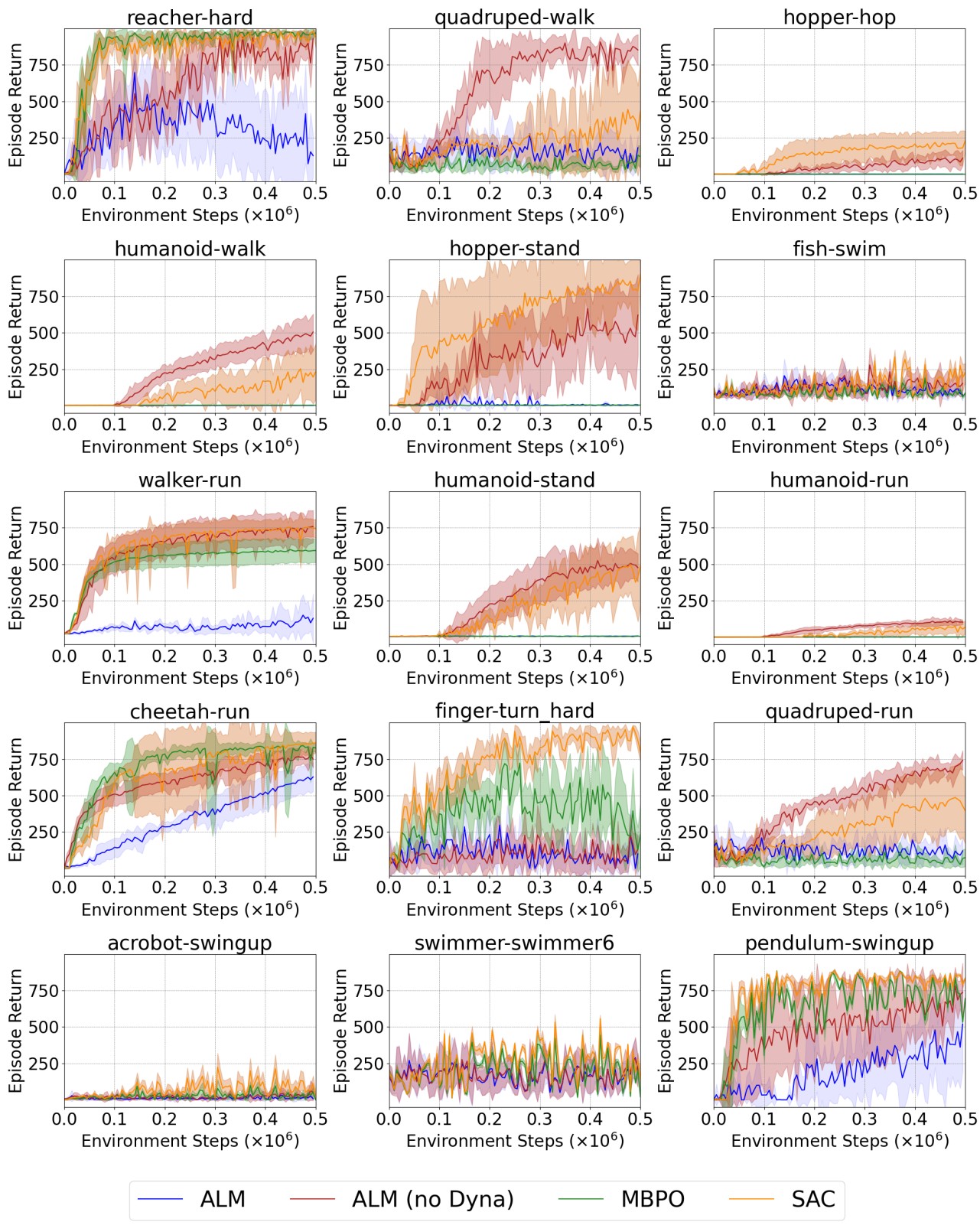

*Figure 17.* Comparing performance of MBPO, SAC, ALM, and ALM without Dyna for 15 challenging DMC benchmark tasks.

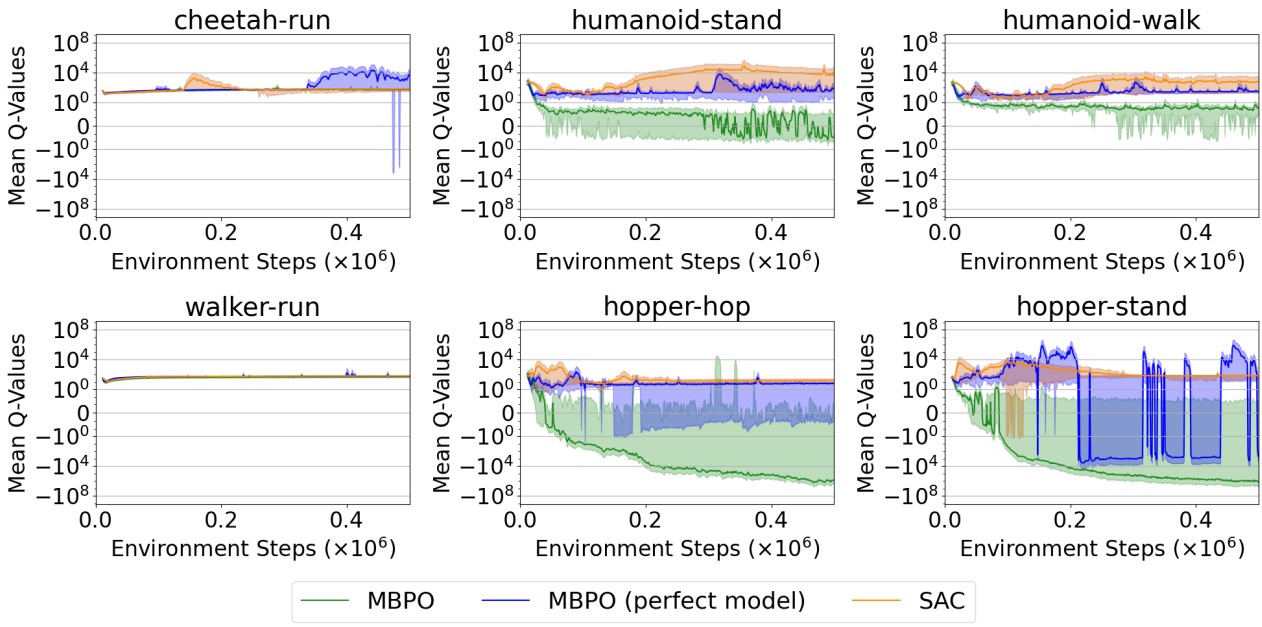

*Figure 18.* Mean Q-values across six seeds each for 6 challenging DMC benchmark tasks. Comparisons are of MBPO and SAC to MBPO with a perfect predictive model as discussed in Section 4.3. Shaded regions correspond to minimum and maximum Q values across trials and solid lines are the mean.

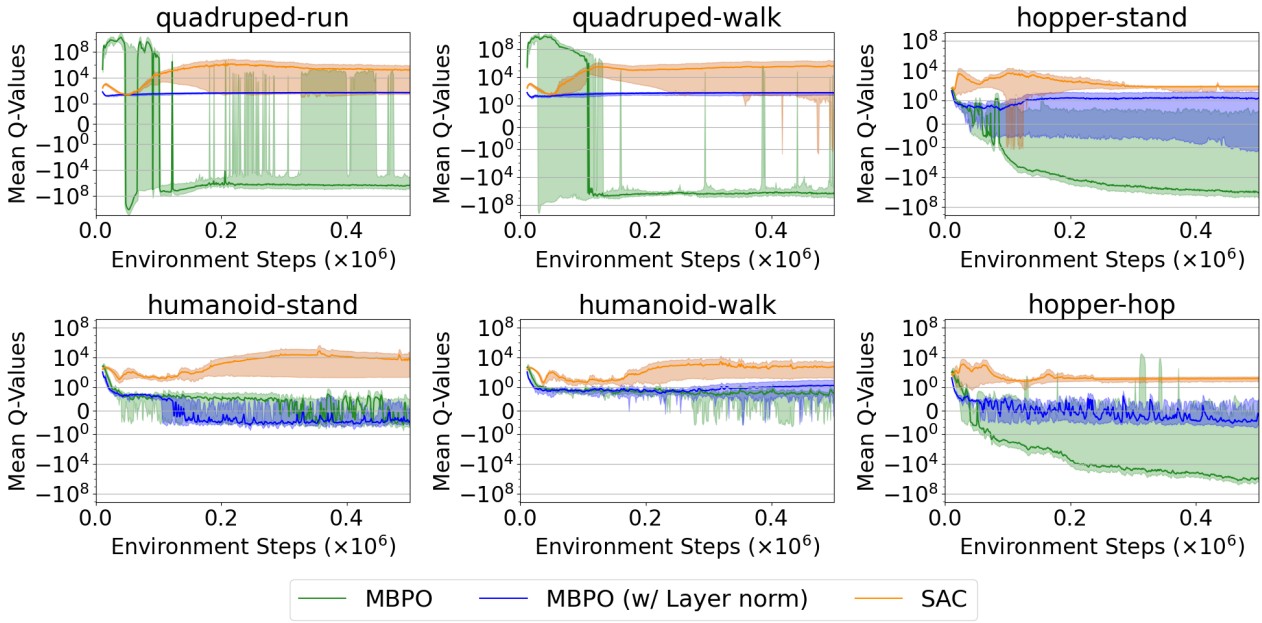

*Figure 19.* Mean Q-values for 6 challenging DMC benchmark tasks across six seeds each. Comparisons are of MBPO and SAC to MBPO with layer norm applied as discussed in Section 4.3. Shaded regions correspond to minimum and maximum Q values across trials and solid lines are the mean.

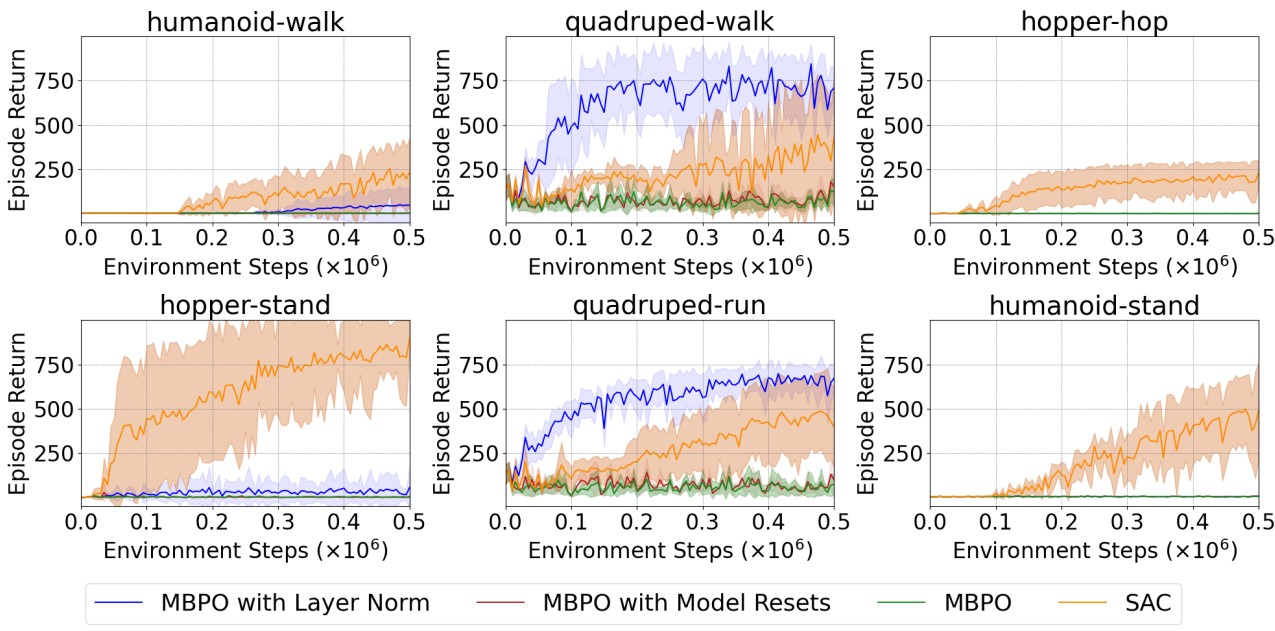

*Figure 20.* Comparing performance of MBPO, SAC, MBPO with layer norm applied to the critic, and MBPO with periodic model resets for 6 challenging DMC benchmark tasks as discussed in Section 5.

