# OpenReview forum: "Stealing That Free Lunch: Exposing the Limits of Dyna-Style Reinforcement Learning"
_ICML.cc/2025/Conference — ICML 2025 poster_

### Official Review · Reviewer_n69W · 2025-02-24

**Overall Recommendation:** 3

**Summary:**

The paper investigates the performance discrepancies of Dyna-style model-based reinforcement learning (DMBRL) algorithms across different benchmark environments, specifically OpenAI Gym and DeepMind Control Suite (DMC). The authors highlight a significant performance gap, where DMBRL algorithms like Model-Based Policy Optimization (MBPO) and Aligned Latent Models (ALM) perform well in OpenAI Gym but struggle in DMC, despite the environments sharing similar tasks and physics backends.

Main Findings:

Performance Gap: DMBRL algorithms show a performance drop in DMC compared to OpenAI Gym.

Model Error Impact: High model error in DMC environments leads to significant performance degradation, even with short synthetic rollouts.

Critic Divergence: Inaccurate synthetic transitions cause critic divergence, hindering learning in DMC.

Main Results:

Empirical Evaluations: MBPO and ALM were evaluated across six OpenAI Gym tasks and fifteen DMC tasks, showing that Dyna-style enhancements can prevent policy improvements in DMC.

Model Error Analysis: Even with a perfect model, MBPO cannot consistently outperform its base off-policy method (SAC) in DMC.

Critic Divergence Mitigation: Layer Normalization mitigates critic divergence but does not enable MBPO to outperform SAC consistently.

Plasticity Loss Mitigation: Periodic resets of off-policy base components improve MBPO's performance in some DMC tasks but do not match SAC's performance.

Contributions:

Identifying Performance Gap: The paper demonstrates a significant performance gap of DMBRL methods across different benchmarks.

Analyzing Causes: The authors analyze potential causes for the performance discrepancy and apply modern mitigation approaches.

Accelerating Experimentation: The paper introduces a new JAX-based implementation of MBPO, achieving up to a 40x decrease in wall-clock time, enabling more accessible DMBRL studies.

## update after rebuttal

Thank you for the author's reply. It answered almost all of my questions. I have no further questions.

**Claims And Evidence:**

Yes, Well-Supported Claims:

Performance Gap: DMBRL algorithms like MBPO and ALM perform well in OpenAI Gym but struggle in DMC. This is backed by empirical evaluations across multiple tasks.

Model Error Impact: High model error in DMC leads to performance degradation, supported by detailed measurements and analysis.

Critic Divergence: Inaccurate synthetic transitions cause critic divergence, hindering learning in DMC. Evidence includes Q-value measurements and analysis of synthetic data impact.

Plasticity Loss: Periodic resets of off-policy base components improve MBPO's performance in some DMC tasks, though not universally.

Areas for Further Clarification:

Dyna-Style Enhancements and Plasticity Loss: More detailed analysis could strengthen the claim that Dyna-style enhancements exacerbate plasticity loss.

Intrinsic Performance Gap: The claim that the performance gap is intrinsic to Dyna-style algorithms could benefit from more direct evidence or theoretical analysis.

**Essential References Not Discussed:**

N/A

**Experimental Designs Or Analyses:**

The paper uses sound experimental designs to support its claims:

Comparative Performance Evaluation: Valid comparison of MBPO and ALM across OpenAI Gym and DMC tasks.

Model Error Analysis: Measures model error effectively to show its impact on performance.

Critic Divergence Analysis: Tracks Q-values to demonstrate the effects of synthetic data on learning.

Plasticity Loss Mitigation: Uses periodic resets to assess the impact of plasticity loss on performance.

Perfect Model Experiments: Modifies DMC to use a perfect model, isolating the effect of model accuracy.

**Methods And Evaluation Criteria:**

Proposed Methods:

Empirical Evaluations: The paper evaluates MBPO and ALM across multiple tasks in OpenAI Gym and DeepMind Control Suite (DMC). This comparative approach is appropriate for identifying performance gaps and understanding the robustness of DMBRL algorithms.

Model Error Analysis: The authors measure model error across training distributions in both OpenAI Gym and DMC. This method is relevant for understanding how model accuracy impacts performance.

Critic Divergence Analysis: The paper measures average critic Q-values during training to assess critic divergence. This method helps identify how inaccurate synthetic transitions affect learning.

Plasticity Loss Mitigation: The authors perform periodic resets of the off-policy base components to mitigate plasticity loss. This method is sensible for evaluating the impact of plasticity loss on performance.

Perfect Model Experiments: The paper modifies the DMC simulator to use a perfect model to evaluate the impact of model accuracy on performance. This method provides insights into the intrinsic limitations of Dyna-style algorithms.

Evaluation Criteria:

Benchmark Datasets: The use of OpenAI Gym and DMC as benchmark datasets is appropriate. Both are widely accepted testbeds for reinforcement learning, providing a variety of continuous control tasks with similar physics backends.

Performance Metrics: The paper uses normalized final return and sample efficiency as performance metrics. These metrics are standard in reinforcement learning and provide a clear basis for comparison.

Model Error Metrics: The authors measure model error as a percentage of the training distribution. This metric is relevant for assessing the accuracy of the predictive model.

Critic Divergence Metrics: The paper measures average critic Q-values to assess critic divergence. This metric is appropriate for evaluating the impact of synthetic data on learning.

**Other Comments Or Suggestions:**

N/A

**Other Strengths And Weaknesses:**

N/A

**Questions For Authors:**

Here are key questions for the authors that could clarify important points:

Synthetic Data Bias: How do you ensure synthetic data doesn’t introduce bias into the critic’s learning process? Are there methods to filter or correct this data?

Plasticity Loss Solutions: Besides periodic resets, have you considered other methods to address plasticity loss, such as regularization or dynamic architectures?

Generalizability: Have you tested these algorithms on other benchmarks or real-world applications to see if the performance gap persists?

**Relation To Broader Scientific Literature:**

Key Relations to Prior Work:

Dyna-Style Algorithms:

Prior Work: Based on Sutton (1991), with extensions like MBPO (Janner et al., 2019).

Contribution: Shows performance gaps across OpenAI Gym and DMC, highlighting robustness issues.

MBRL Algorithms:

Prior Work: MBPO and ALM (Ghugare et al., 2023) demonstrate success in specific benchmarks.

Contribution: Reveals degraded performance in DMC, emphasizing the need for diverse benchmarking.

**Theoretical Claims:**

This paper focuses on empirical evaluations rather than theoretical claims, so there are no formal proofs to verify. However, the key conceptual claims are well-supported by empirical evidence:

Dyna-style enhancements can prevent policy improvements in DMC: Supported by performance metrics across multiple tasks.

High model error in DMC leads to performance degradation: Backed by detailed measurements of model error.

Critic divergence due to inaccurate synthetic transitions hinders learning: Evidenced by Q-value measurements and analysis.

Plasticity loss contributes to MBPO's failures: Demonstrated through periodic resets and their impact on performance.

Issues:

Dyna-Style Enhancements and Plasticity Loss: Could use more detailed theoretical analysis.

Intrinsic Performance Gap: Would benefit from additional theoretical insights.

---

> ### Author Rebuttal · Authors · 2025-03-29
>
> **How do you ensure synthetic data doesn’t introduce bias into the critic’s learning process? Are there methods to filter or correct this data?**
>
> We performed sweeps over the synthetic-to-real data ratio (Figure 12) and found that introducing any amount of synthetic data into off-policy learning in DMC leads to a substantial degradation in policy performance. Moreover, in our Dyna-style experiments using perfect synthetic data (i.e., no model error), we showed in Figure 6 that learning still fails to outperform the base off-policy methods on which these algorithms are built. These findings suggest that the failure is not due to model inaccuracies or data quality, but rather due to algorithmic issues intrinsic to how synthetic data is integrated.
>
> While methods such as those proposed in [1,2] attempt to filter or adapt synthetic rollouts, our results imply that such corrections are insufficient to fix the core algorithmic shortcomings. Even with perfect data, the performance gap remains, underscoring the limits of current Dyna-style designs.
>
> Addressing this challenge remains an active area of research for us, and we welcome further discussion on promising directions for improving synthetic data integration in off-policy learning.
>
> **Besides periodic resets, have you considered other methods to address plasticity loss, such as regularization or dynamic architectures?**
> Yes. While we did not frame them explicitly in terms of plasticity, we explored several techniques targeting related issues. Layer normalization was employed in prior work [3] as a method that simultaneously mitigates overestimation and stabilizes learning dynamics - both of which relate to plasticity loss. Additionally, we experimented with unit ball normalization [4],  which was proposed to combat overestimation and thereby also helps address plasticity indirectly. We did not include these results in the main text as they did not substantially outperform the more commonly used layer norm baseline.
>
> We did not pursue further architectural or regularization strategies because our literature review found no alternatives that clearly outperform these methods. Periodic resetting, layer norm, and unit ball normalization appear to represent the state-of-the-art approaches for maintaining plasticity in high-update-ratio settings. If we have overlooked a promising direction, we would sincerely appreciate any further insight from the reviewer.
>
> **Have you tested these algorithms on other benchmarks or real-world applications to see if the performance gap persists?**
> We did not. Our motivation was to critically re-examine claims of sample efficiency made by prior dyna-style methods (e.g., ALM, MBPO), which were accepted largely based on Gym benchmarks. These methods were widely adopted under the assumption that they outperformed their model-free counterparts in sample efficiency. Our work revisits that conclusion using a comparable benchmark - DeepMind Control Suite (DMC) - which shares the same MuJoCo physics backend.
>
> This choice minimizes the likelihood that differences in performance are due to changes in dynamics simulation, and therefore emphasizes the impact of the Dyna-based algorithmic changes introduced in ALM and MBPO. Our results demonstrate that the original claims made about those methods do not generalize even within this extremely similar experimental setting. While evaluating additional domains might provide further insight into generalization challenges, we chose to focus on a clean and controlled comparison to highlight foundational flaws in current Dyna-style designs. Introducing more benchmarks could risk diluting the clarity of this central message.
>
> **Citations:**
>
> [1] B. Frauenknecht, A. Eisele, D. Subhasish, F. Solowjow, and S. Trimpe, “Trust the Model Where It Trusts Itself -- Model-Based Actor-Critic with Uncertainty-Aware Rollout Adaption,” Jun. 21, 2024, arXiv: arXiv:2405.19014. doi: 10.48550/arXiv.2405.19014.
>
> [2] Y. Li, Z. Dong, E. Luo, Y. Wu, S. Wu, and S. Han, “When to Trust Your Data: Enhancing Dyna-Style Model-Based Reinforcement Learning With Data Filter,” Oct. 16, 2024, arXiv: arXiv:2410.12160. doi: 10.48550/arXiv.2410.12160.
>
> [3] C. Lyle, Z. Zheng, E. Nikishin, B. A. Pires, R. Pascanu, and W. Dabney, “Understanding plasticity in neural networks,” Nov. 27, 2023, arXiv: arXiv:2303.01486. Accessed: Jun. 29, 2024. [Online]. Available: http://arxiv.org/abs/2303.01486
>
> [4] M. Hussing, C. Voelcker, I. Gilitschenski, A. Farahmand, and E. Eaton, “Dissecting Deep RL with High Update Ratios: Combatting Value Overestimation and Divergence,” Mar. 09, 2024, arXiv: arXiv:2403.05996. Accessed: Apr. 08, 2024. [Online]. Available: http://arxiv.org/abs/2403.05996

---

### Official Review · Reviewer_ffFz · 2025-03-06

**Overall Recommendation:** 4

**Summary:**

This paper investigates why two MBRL algorithms (MBPO & ALM) perform well on OpenAI Gym but poorly on DMC.
The authors show that apart from Dyna model prediction error, the synthetic rollouts themself could arrest policy improvement rather than enhance it when deployed across more diverse environments.

**Claims And Evidence:**

The claims made in the submission are mostly supported by clear and convincing evidence

**Essential References Not Discussed:**

No missing key references found yet.

**Experimental Designs Or Analyses:**

The soundness/validity of the experimental designs or analyses is good if the implementation is correct.

**Methods And Evaluation Criteria:**

The methods used mostly fit the problem and the authors' claims.

**Other Comments Or Suggestions:**

1. At the abstract/intro of this paper, I think it would be better to say something like "$\underline{\text{control tasks}}$ in OpenAI Gym", since it contains environments other than control tasks.

2. On the left column of the line 057, I suggest cite ALM when it first appears and reduce other repeated citations later.

**Other Strengths And Weaknesses:**

**Strengths:**

1. There are many papers investigating new MBRL models, but few studies the underlying reason why modern MBRL works, or why a learned model could improve the sample efficiency of model-free RL algorithms. Thus the community lacks relevant theories for design. This paper is a good start in bridging this gap.

2. The authors' claims are clear and the evidence is supportive. They studied the impact of model error, underestimation, learning dynamics and so on, so that the limitation of the Dyna style's learning is clear.

**Weaknesses:**

See comments and questions.

**Questions For Authors:**

1. What would be the result of Figure 12 if the model is perfect?

2. For MBPO/ALM on OpenAI Gym that surpasses SAC (or like DreamerV3/EfficientZeroV2 on DMC), could you also observe the underestimation issue? This may provide insights for further improving the SOTA MBRL methods.

3. For the left column of the line 425. When resetting the predictive model, it may take more updates than the original train ratio to optimize the model. If the number of updates is too low, then the model may fail to converge and provide little information. If the number of updates is sufficient, then the key insights 1 may still hold. Could you clarify the details on this?

**Relation To Broader Scientific Literature:**

This paper studies the problem that the performance of MBRL algorithms varies depending on the environment and problem characteristics. Some previous work pointed out the inefficiency of MBRL on DMC tasks, and some previous work studied the impact of model error.

**Theoretical Claims:**

Few theoretical proofs are involved in this paper.

---

> ### Author Rebuttal · Authors · 2025-03-29
>
> We thank the reviewer for their helpful suggestions to improve the clarity and presentation of our paper. We will incorporate the suggested changes to:
> - More precisely describe the subset of OpenAI Gym environments used in our experiments.
> - Cite ALM upon first mention and reduce repeated citations later on to avoid redundancy.
>
> **What would be the result of Figure 12 if the model is perfect?**
> This is an excellent question and one we are also curious about. As depicted in Figure 6, we generated a single data point for both humanoid-stand and hopper-stand at the 0.05 synthetic ratio using perfect transitions (i.e., querying the ground truth environment). This process required approximately 4 GPU-days and substantial CPU usage, due to the environment transitions becoming the bottleneck. Unlike synthetic rollouts from learned models (which are efficiently parallelizable via JAX), real environment rollouts suffer from parallelism limitations that drastically slow down runtime.
>
> Extrapolating from our timing, generating the full version of Figure 12 with perfect transitions would require roughly 40 GPU-days across multiple machines. Given the resource cost and limited additional insight expected beyond what is already shown in the paper, we opted not to pursue the full experiment at this time. However, if we succeed in optimizing environment rollouts via vectorized environments in the future, we agree this would be a worthwhile avenue to explore.
>
> That said, our preliminary results suggest that even with perfect transitions, performance does not recover to the level of pure real-data training, regardless of synthetic-to-real ratio. This aligns with our central claim that the degradation arises not solely from model bias but from algorithmic limitations in how synthetic data is incorporated.
>
> **For MBPO/ALM on OpenAI Gym that surpass SAC (or methods like DreamerV3/EfficientZeroV2 on DMC), do you also observe the underestimation issue?**
> Thank you for this insightful question. We interpret it as: "Do value underestimation issues persist in model-based methods when they outperform SAC in Gym or state-of-the-art model-free methods in DMC?" Please let us know if we have misunderstood.
>
> In our experiments, we did not observe value underestimation in ALM or MBPO when evaluated on OpenAI Gym. This suggests that underestimation is largely a DMC-specific phenomenon in our setting. As discussed in the paper, even with a perfect model, underestimation is much less pronounced in DMC compared to a learned model, implicating model error as a contributor to Dyna failing, but not the only issue. However, Section 4.2 further shows that MBPO still fails to match SAC in DMC even when underestimation is not prominent, indicating that underestimation, while influential, is not the sole factor - nor a "smoking gun" - behind the performance gap.
>
> **For the left column of line 425 - could you clarify the update count after model reset?**
> Certainly. During our investigation of model plasticity, we explored multiple reset strategies inspired by prior work [1]. We found that increasing the reset frequency up to a point improved performance. The strategy we ultimately adopted - resetting every 20k environment steps - aligns with the reset frequency recommended in [1] for a replay ratio of 128. Empirically, this approach worked better in our setting than the more conservative reset frequency from [1] prescribed for a replay ratio of 20 (every 2.56 × 10⁶ gradient steps, equivalent to ~128k env steps), or intermediate strategies between the two.
>
> We will clarify this detail in the revision to aid reproducibility and interpretability.
>
> Citations:
> [1] P. D’Oro, M. Schwarzer, E. Nikishin, P.-L. Bacon, M. G. Bellemare, and A. Courville, “Sample-Efficient Reinforcement Learning by Breaking the Replay Ratio Barrier,” 2023.

---

> > ### Comment · Reviewer_ffFz · 2025-04-03
> >
> > Thank you for the detailed response, most of my concerns are resolved. I'll keep my score as it is.

---

### Official Review · Reviewer_TyLB · 2025-03-13

**Overall Recommendation:** 2

**Summary:**

This paper shows that Dyna-style off-policy model-based reinforcement learning (DMBRL) algorithms perform well in OpenAI Gym, while their performance can drop significantly in DeepMind Control Suite (DMC). And the paper analyzes potential causes (model error, lay normalization, etc) for this discrepancy, which ultimately fail to consistently resolve these problems.

**Claims And Evidence:**

Yes.

**Essential References Not Discussed:**

The paper could benefit from discussing PlaNet (Hafner et al., 2019) and PETS (Chua et al., 2018), which are other model-based RL algorithms that have been evaluated in both Gym and DMC. Additionally, the paper could discuss more recent advancements in model-based RL. This could provide a broader perspective on the limitations of purely Dyna-style methods.

**Experimental Designs Or Analyses:**

Yes.

**Methods And Evaluation Criteria:**

Yes.

**Other Comments Or Suggestions:**

1. **Clarification on Model Error Metrics:** The paper uses "percent model error" as a metric, but it is not clearly defined how this metric is calculated. Providing a precise definition or formula for this metric would improve the clarity of the results.

2. **Reproducibility:** While the authors claim a JAX-based implementation, it would be helpful to provide the code details.

**Other Strengths And Weaknesses:**

**Strengths:**

1. The paper presents a observation that Dyna-style algorithms, which perform well in OpenAI Gym, significantly underperform in DeepMind Control Suite (DMC). This finding challenges the assumption that these algorithms generalize well across similar benchmarks and raises important questions about the robustness of DMBRL methods.

2. The paper provides a thorough empirical analysis, including extensive experiments across multiple environments and benchmarks. The authors systematically investigate potential causes for the performance gap, such as model error, overestimation bias, and neural network plasticity, and provide detailed results to support their conclusions.

**Weaknesses:**

1. While the paper focuses on MBPO and ALM, it does not sufficiently explore other model-based RL algorithms that have shown success in DMC, such as DreamerV3. A more comprehensive comparison with these algorithms would provide a clearer picture of why certain model-based methods succeed where others fail.

2. The paper briefly mentions differences between OpenAI Gym and DMC in terms of reward structures, termination conditions, and physical parameters, but it does not delve deeply into how these differences might contribute to the performance gap. A more detailed analysis of these environmental factors could provide additional insights into the limitations of Dyna-style algorithms.

3. The paper mentions hyperparameter sweeps but does not provide a detailed analysis of how sensitive the results are to different hyperparameter settings. Given that hyperparameters can significantly impact the performance of RL algorithms, a more thorough exploration of this aspect would strengthen the paper's conclusions.

**Questions For Authors:**

1. The paper suggests that the performance gap is "fundamental," but have the authors considered the possibility that it could be mitigated through more sophisticated model architectures or training techniques, such as meta-learning or transfer learning?

2. Given that DreamerV3 performs well, what specific design choices or architectural differences might explain its success compared to MBPO and ALM?

**Relation To Broader Scientific Literature:**

The paper shows a significant gap in the literature regarding the performance of Dyna-style model-based reinforcement learning (DMBRL) algorithms across different benchmarks, specifically OpenAI Gym and DeepMind Control Suite (DMC). The findings challenge the prevailing assumption that DMBRL algorithms, such as Model-Based Policy Optimization (MBPO), generalize well across similar environments.

**Theoretical Claims:**

The paper does not contain proofs or theoretical developments.

---

> ### Author Rebuttal · Authors · 2025-03-29
>
> **On comparisons with other model-based RL algorithms:**
> We appreciate the reviewer’s suggestion regarding DreamerV3, PlaNet, and PETS. Our study focuses on Dyna-style algorithms MBPO and ALM, which use synthetic rollouts in proprioceptive, state-based settings. Pixel-based methods such as DreamerV1-V3 and PlaNet abstract away low-level issues central to our study. From the perspective of pixels, two hopper environments that move and look similar are effectively indistinguishable, whether simulated in Gym or DMC. In contrast, state-based methods must engage with environment-specific dynamics and state representations, which our results show can vary significantly despite a shared MuJoCo backend. For this reason, pixel-based approaches fall outside the scope of our analysis.
>
> DreamerV3, although impressive on proprioceptive tasks in addition to pixel-based, significantly differs from simpler Dyna-style methods in complexity and design philosophy. Understanding its success relative to MBPO and ALM, as noted in Section 2.1 and our FAQ, is valuable but warrants a dedicated analysis. Additionally, DreamerV3 has not been evaluated on OpenAI Gym tasks, complicating comparisons due to benchmarking inconsistency (as discussed in our paper).
>
> Similarly, PETS and TD-MPC diverge from the Dyna paradigm by employing learned models for online trajectory optimization (e.g., model predictive control), rather than using synthetic rollouts for policy/value training. Our work targets hallucinated experience in Dyna-style pipelines.
>
> Ultimately, while broader comparisons are important, evaluating methods that differ in observation modality or depart from the Dyna structure falls outside our intended scope. Our goal is to identify previously overlooked failure modes in a prominent subclass of model-based RL - Dyna-style methods - under realistic, proprioceptive conditions.
>
> **On the possibility that more sophisticated models could close the performance gap:**
> We thank the reviewer for the insightful question regarding whether more sophisticated model architectures or training techniques (e.g., meta-learning) could close the performance gap. While such advances may indeed improve results, they would only reinforce our central claim: standard Dyna-style methods like MBPO and ALM exhibit fundamental limitations when used “out of the box,” even with extensive tuning. These methods were originally presented as simple, sample-efficient augmentations to off-policy RL, relying on short synthetic rollouts and standard neural network dynamics models. Yet across 15 diverse DMC environments, they consistently underperform their base off-policy RL learners - and sometimes even random policies - revealing deeper algorithmic issues.
>
> **On environmental differences between DMC and Gym:**
> We appreciate the reviewer’s point that differences in reward structures, termination conditions, and dynamics between OpenAI Gym and DMC may influence performance. While we address these factors (Section 2.2, Appendix C), our analysis in the paper is deliberately algorithm-focused. We show that the base algorithm, SAC, performs well in both testbeds, while its Dyna-style counterpart (e.g., MBPO) fails in DMC. This contrast isolates synthetic rollouts - not environment differences - as the primary source of degradation, challenging the assumption that Dyna-style methods reliably enhance strong off-policy learners across domains. We agree that further work should explore how environment-side factors interact with Dyna-style methods. These are important questions, but beyond the current paper’s scope.
>
> **On hyperparameter sensitivity:**
> As described at the end of Section 4.1, we performed extensive hyperparameter sweeps on key environments (e.g., hopper-stand, humanoid-stand), varying model size, retraining frequency, learning rates, and gradient steps. While these reduced model error in some cases, they did not produce consistent or meaningful gains in policy performance beyond a random policy in many cases. Sweeps over the off-policy algorithm's hyperparameters yielded similarly negligible effects.
> This insensitivity supports our central claim: the failure of Dyna-style methods arises from fundamental challenges in leveraging synthetic rollouts, not suboptimal hyperparameter choices. We therefore did not emphasize sensitivity analyses, as tuning alone does not address the core limitations of these approaches. We are happy to include additional details in the camera-ready manuscript.
>
> **Clarification on the “percent model error” metric:**
> We will revise the text to include an explicit formula and explanation of our percent model error metric.
> Specifically, let $\hat{y}$ denote the model's prediction (next observation and reward) and let $y$ be the corresponding ground-truth target. We define:
>
> $$
> \text{Percent Model Error} =  \frac{ \left\| \hat{y} - y \right\|_2 }{ \left\| y \right\|_2 } \times 100
> $$
>
> **Reproducibility:**
>
> See response to Reviewer gMX1.

---

> > ### Comment · Reviewer_TyLB · 2025-04-08
> >
> > Thank authors for a detailed response to my review.
> >
> > The rebuttal argues that sophisticated techniques (e.g., meta-learning) would merely "reinforce" their claim of Dyna’s limitations. However, this reasoning conflates algorithmic flaws with practical deployability:
> > - If advanced methods (e.g., uncertainty-aware rollouts or hierarchical models) could mitigate the issues, the problem lies in engineering, not theory.
> > - The paper’s sweeping conclusion—"no free lunch"—loses weight without testing whether the "lunch" could be improved with better ingredients (e.g., robust models).
> >
> > The rebuttal mentions "extensive" sweeps. Could the authors release full sweep data (e.g., in supplementary materials) to demonstrate its claim?

---

> > > ### Author Response · Authors · 2025-04-09
> > >
> > > We respect the philosophical distinction the reviewer highlights, and agree that separating theoretical soundness from practical implementation is important. However, in deep RL, theoretical assumptions are often quickly challenged once deep neural networks and complex continuous-control tasks are introduced. In this context, the performance of an algorithm becomes closely tied to its engineering choices.
> > >
> > > To reiterate our position, our claim is not that Dyna-style methods are theoretically unsound and we explicitly state this in Appendix A, FAQ 1, Section 2.1, and in our original rebuttal. Rather, our work highlights that widely used practical instantiations of Dyna-style algorithms, specifically those using short synthetic rollouts with simple MLP dynamics models, are brittle across realistic benchmarks. Importantly, methods like MBPO and ALM were not presented as algorithms tailored specifically to OpenAI Gym. They were positioned as general-purpose solutions for continuous control, and are widely cited and adopted as such. That they fail to generalize beyond Gym, even after extensive hyperparameter sweeps (which we will include in the camera-ready appendix), suggests that the issue is not one of tuning, but of fundamental limitations in their standard design. While more sophisticated techniques (e.g., uncertainty-aware rollouts, hierarchical models, meta-learning) might improve performance, the need for such non-trivial modifications constitute entire new publications and only reinforces our core point: the promise of “plug-and-play” model-based RL - as conveyed by methods like MBPO and ALM - is undermined if success depends on significant architectural or procedural overhauls. We will revise the language in the introduction and conclusion to clarify that our critique is aimed at the promise of standard Dyna-style algorithms (i.e,. MBPO and ALM), and not at the theoretical soundness of the broader Dyna framework.
> > >
> > > To further clarify, we included DreamerV3 as an example of a Dyna-style method that achieves success, but only after years of sustained engineering effort. Its impressive performance, in our view, reinforces rather than contradicts our claim.
> > >
> > > Thus, our “no free lunch” conclusion is not a rejection of model-based RL, but a caution: several popular Dyna-style methods, in their standard forms, lack the robustness and generality often assumed based on their published results. If improving the “lunch” requires entirely new ingredients, then it is no longer the same recipe.

---

### Official Review · Reviewer_gMX1 · 2025-03-14

**Overall Recommendation:** 4

**Summary:**

The paper investigates why popular Dyna‐style model‐based reinforcement learning (RL) methods, such as MBPO and ALM, perform well on OpenAI Gym tasks but struggle on the DeepMind Control Suite (DMC), despite both benchmarks having similar physics and task structures. The authors document a consistent performance gap: when synthetic rollouts—core to the Dyna approach—are used, these methods can even underperform a random policy in DMC, while in Gym they improve sample efficiency over their model‐free counterparts like SAC. The key contributions include:
- Demonstrating that Dyna-style enhancements, which generate synthetic transitions to supplement real experience, can impede policy improvement in DMC environments even when the underlying off-policy algorithm works well on its own.
- Conducting extensive experiments to analyze potential causes such as high model error, critic divergence (stemming from over- or underestimation due to unrealistic synthetic data), and issues with network plasticity. The study shows that even using a perfect predictive model or applying techniques to stabilize critic updates does not fully close the performance gap.
- Introducing a new JAX-based implementation that accelerates experimentation by up to 40× compared to previous PyTorch-based approaches, thereby reducing the computational barrier for thorough evaluation of model-based RL algorithms.

**Claims And Evidence:**

The assertion that Dyna-style enhancements consistently hinder performance in DMC is well illustrated by experiments with MBPO and ALM, yet the evidence is drawn from a specific set of environments and methods. While the paper argues these issues are inherent to the Dyna family, the existence of approaches like DreamerV3 (which succeed in DMC) suggests that further investigation is needed to determine if the observed limitations are truly universal across all Dyna-style methods.

**Essential References Not Discussed:**

N/A

**Experimental Designs Or Analyses:**

The experimental design and analyses are sound, with thoughtful controls and comparisons.
The paper rigorously compares Dyna-style methods (MBPO and ALM) against their base off-policy counterparts (e.g., SAC) across two distinct benchmark suites—OpenAI Gym and the DeepMind Control Suite (DMC). This dual-benchmark evaluation is a strong point because it highlights the performance discrepancies in different settings. The use of controlled ablation studies (e.g., varying the synthetic-to-real data ratio, using a perfect predictive model, and applying periodic resets) provides granular insights into potential sources of failure such as critic divergence and network plasticity. This detailed analysis helps isolate the factors contributing to the performance gap.

**Methods And Evaluation Criteria:**

The paper’s methodology is well-aligned with its goal of probing the limits of Dyna-style RL. The authors use two widely recognized benchmarks—OpenAI Gym and DeepMind Control Suite—to highlight that improvements seen with synthetic rollouts in one setting do not necessarily carry over to another with similar physics but subtle differences in task dynamics. The paper further dissects key factors (e.g., model error, critic divergence, and network plasticity) through controlled experiments and ablation studies.

**Other Comments Or Suggestions:**

It would be great if code is available for this work.

**Other Strengths And Weaknesses:**

The work creatively challenges established successes of Dyna-style methods by systematically comparing performance across benchmarks that, on the surface, appear similar. This cross-benchmark analysis exposes hidden limitations, an angle that is relatively novel in the literature.  By demonstrating that enhancements such as synthetic rollouts can actually impede performance in certain environments, the study raises important questions about the generality and robustness of widely adopted model-based RL methods.

**Questions For Authors:**

I have no questions

**Relation To Broader Scientific Literature:**

The paper revisits and challenges the celebrated gains of MBPO (Janner et al., 2019) and similar Dyna‐style methods, which had previously been shown to dramatically improve sample efficiency in OpenAI Gym. By exposing a performance gap when these techniques are applied to the DeepMind Control Suite (DMC), the paper questions the generality of prior successes.

**Theoretical Claims:**

The paper invokes the “no free lunch” concept to argue that no single RL algorithm can be optimal across all environments, but it does not provide a formal proof of this claim—it leans on established theoretical principles and empirical observations.

---

> ### Author Rebuttal · Authors · 2025-03-29
>
> Thank you for your thoughtful and positive review. We appreciate your recognition of our contributions and the strengths of our experimental design and analysis.
>
> Regarding your mention of DreamerV3, we agree that its success in DMC highlights the diversity of model-based RL approaches. For further detail, we refer the reviewer to the section titled "On comparisons with other model-based RL algorithms" in our response to Reviewer TyLB, clarifying why DreamerV3 falls outside the scope of our evaluation of Dyna-style methods.
>
> Regarding the code, we consider the code an integral part of our contributions and are actively refining it. If the paper is accepted, we will certainly release the code alongside the camera-ready version.

---

### Decision · Program_Chairs · 2025-05-01

**Decision:**

Accept (poster)

**Comment:**

This paper investigates Dyna-like reinforcement learning algorithms and, especially, addresses the differences in performance on standard benchmarks (OpenAI Gym and DeepMind Control Suite).

Their is a consensus that the problem is important and is worth studying. Most of the reviewers appreciated the experimental setup and the rigorous methodology that challenges the different dimensions of Dyna-like RL methods. Especially, the generation of synthetic roll outs that seems to harm performance in some contexts while it enhances performance in others.

Among the raised concerns, the choice of the tested algorithms was pointed as a weakness of the paper. Especially, no comparisons to sophisticated methods such as DreamerV3 was made. It makes the paper somewhat lacking of completeness and the authors were encouraged to study such methods.